# On Uniform, Bayesian, and PAC-Bayesian Deep Ensembles

## Abstract

It is common practice to combine deep neural networks into ensembles. These deep ensembles can benefit from the cancellation of errors effect: Errors by ensemble members may average out, leading to better generalization performance than each individual network. Bayesian neural networks learn a posterior distribution over model parameters, and sampling and weighting networks according to this posterior yields an ensemble model referred to as a Bayes ensemble. In this study, we review the argument that neither the sampling nor the weighting in Bayes ensembles are particularly well suited for increasing generalization performance, as they do not support the cancellation of errors effect, which is evident in the limit from the Bernstein-von Mises theorem for misspecified models. In contrast, we show that a weighted average of models, where the weights are optimized by minimizing a second-order PAC-Bayesian generalization bound, can improve generalization performance. It is crucial that the optimization takes correlations between models into account. This can be achieved by minimizing the tandem loss, which requires hold-out data for estimating error correlations. The tandem loss based PAC-Bayesian weighting increases robustness against correlated models and models with lower performance in an ensemble. This allows us to safely add several models from the same learning process to an ensemble, instead of using early-stopping for selecting a single weight configuration. Our study presents empirical results supporting these conceptual considerations on four different classification datasets. Our experiments provide further evidence that state-of-the-art intricate Bayes ensembles from the literature do not outperform simple uniformly weighted deep ensembles in terms of classification accuracy. Additionally, we show that these Bayes ensembles cannot match the performance of deep ensembles weighted by optimizing the tandem loss, which additionally provides nonvacuous rigorous generalization guarantees.

## 1 Introduction

Combining different deep neural networks into an ensemble model is a common way to increase generalization performance (Bishop & Bishop, 2023; Goodfellow et al., 2016). Such deep ensembles can profit from the cancellation of errors effect (Eckhardt & Lee, 1985): When the individual networks perform better than random guessing and make independent errors, the errors in an additive ensemble (or committee) average out, and the ensemble tends to outperform the individual models (Hansen & Salamon, 1990; Perrone & Cooper, 1993; Dietterich, 2000). Many strategies have been explored to create networks for an ensemble (e.g. Buschjäger et al., 2020; D'Angelo & Fortuin, 2021; Dziugaite & Roy, 2017; Garipov et al., 2018; Huang et al., 2017; Jiang et al., 2017; Lee et al., 2015; Liu & Yao, 1999; Masegosa et al., 2020; Monteith et al., 2011; Ortega et al., 2022; Pérez-Ortiz et al., 2021). However, simple deep ensembles, which rely solely on repeating neural network training with random weight initializations and stochastic training to generate ensemble members, often already improve generalization over single networks (Lakshminarayanan et al., 2017).

One way to build a neural network ensemble is based on the predictive posterior of Bayesian neural networks (Adlam et al., 2020; Aitchison, 2021; Chen et al., 2014; Farquhar et al., 2020; Gal & Ghahramani, 2016; Grünwald, 2012; Izmailov et al., 2021; Kapoor et al., 2022; Kendall & Gal, 2017; MacKay, 1992b; Maddox et al., 2019; Monteith et al., 2011; Nabarro et al., 2022; Neal, 1992; 1996; 2011; Pearce et al., 2020; Ritter et al., 2018; Welling & Teh, 2011; Wenzel et al., 2020a;b; Wilson & Izmailov, 2020; Zhang et al., 2020), and

this approach has recently gained attention (Bachmann et al., 2022; Fortuin et al., 2022; Rudner et al., 2022; Wiese et al., 2023). A Bayesian neural network learns a posterior distribution over the weights of the network (MacKay, 1992a; Neal, 1996). Averaging networks sampled from this posterior yields an approximation of the Bayesian model average (BMA) and is referred to as a Bayes ensemble. This approach comes with great expectations, for example, Domingos (2000) stated 'Given the "correct" model space and prior distribution, Bayesian model averaging is the optimal method for making predictions; in other words, no other approach can consistently achieve lower error rates than it does.' While Bayes ensembles typically outperform individual ensemble members, *we argue that neither the sampling nor the weighting in a Bayes ensemble are particularly well-suited for increasing generalization performance in theory and practice compared to other ensembling methods* (see also Minka, 2002; Ortega et al., 2022). The Bernstein-von Mises theorem (assuming identifiability of the model) shows that the BMA converges towards the maximum likelihood point estimate with growing dataset size (Kleijn & Van der Vaart, 2012). In the limit, it will eventually concentrate on a single model without exploiting ensemble diversity. Thus, the Bayes ensemble typically does not leverage the cancellation of errors effect. This is particularly problematic because the model space considered in the BMA setting is typically not 'correct': The Bayes ensemble itself is, in general, outside the model space from which its members are sampled – and in the space of ensemble models the Bayes ensemble cannot be expected to be optimal.

PAC-Bayesian methods (McAllester, 1998) provide an alternative approach to incorporate prior information about models and to improve the weighting of the members of the ensemble. We bring forward optimizing the weighting of deep ensemble members using a PAC-generalization bound based on the tandem loss, which accounts for pairwise correlations between networks (Masegosa et al., 2020). This allows us to maintain diversity and increase the generalization performance of the deep ensemble. This weight optimization is especially useful when intermediate weight configurations (snapshots) from model training are considered, in which case the optimization performs model selection taking both individual network performance and ensemble diversity into account. The price to pay for the improved weighting is additional hold-out data for the optimization of the weights; however, the resulting PAC-Bayesian bound provides a rigorous performance guarantee.

Our study transfers existing, but scattered and often partly neglected knowledge on the (1) behavior of Bayesian model averaging (BMA), in particular its suboptimality in terms of generalization performance in the (typical) misspecified setting, (2) cancellation of errors, (3) and second-order PAC-Bayesian bounds to ensembling of deep neural networks (DNNs). The main contributions can be summarized as:

- We discuss the conceptual differences between simple, Bayes and PAC-Bayesian deep ensembles, in particular reviewing the argument that Bayes ensembles do not consider the cancellation of errors effect, which may explain some observations in the current literature (e.g., on "cold posteriors").

- We support our conceptual arguments through unbiased experiments on DNN benchmark problems that were used to promote BMA. Our results on four datasets provide further evidence that complex Bayesian approximate inference methods can often be surpassed by more efficient simple deep ensembles. Bagging decreased predictive performance, showing the trade-off between the degree of randomization and single-network performance given limited training data.

- We show that the optimization of a PAC-Bayesian generalization bound using the tandem loss can improve the predictive performance of deep ensembles and provides nonvacuous generalization guarantees. This is in contrast to what could have been expected from the results by Ortega et al. (2022). Our results stress the importance of *second-order* PAC-Bayesian analysis for ensembles.

- We demonstrate that when intermediate checkpoints from the same training run are used in a neural network ensemble, optimizing their weighting by minimizing a second-order PAC-Bayesian bound can improve accuracy and prevent checkpoints from low-performing models from deteriorating the ensemble generalization performance.

In the next section, we will briefly summarize the background on Bayesian model averaging, the cancellation of errors effect, deep ensembles, and PAC-Bayesian majority voting. Section 3 will bring these topics together:

It contrasts uniform, Bayes, and PAC-Bayesian deep ensembles and formulates our main claims. These claims will be empirically studied in Section 4 before we conclude.

## 2 Background

We consider data $\mathcal{D} = \{(x_i, y_i)\}_{i=1}^n$ drawn i.i.d. from a distribution $p$. The loss of a hypothesis/model $h$ on $(x, y)$ is denoted by $\ell(h(x), y)$. The generalization error (risk) of $h$ is given by $L(h) = \mathbb{E}_{(x,y) \sim p}[\ell(h(x), y)]$ and the empirical risk by $\hat{L}_{\mathcal{D}}(h) = \frac{1}{n} \sum_{i=1}^n \ell(h(x_i), y_i)$. We write $h_w$ to stress that a model is parameterized by $w \in \mathbb{R}^d$. We consider ensembles of $M$ models $h_i = h_{w_i} \in \mathcal{H}$, $i = 1, \ldots, M$, weighted by $\rho \in \mathbb{R}^M$. For regression, the ensemble regressor is $h_\rho(x|h_1, \ldots, h_M) = \sum_{i=1}^M \rho_i h_i(x)$. For classification, we distinguish two ways of combining the predictions of the ensemble members. We assume a predictive distribution $p_i(y|x) = p(y|x, w_i)$ associated with each hypothesis $h_{w_i}(x)$. The ensemble prediction $h_\rho(x|h_1, \ldots, h_M)$ can be either defined by the $\rho$-weighted average of these distributions

$$\text{AVG}_\rho(x) = \arg\max_y \sum_{i=1}^M \rho_i p_i(y|x) \tag{1}$$

or by the $\rho$-weighted majority vote

$$\text{MV}_\rho(x) = \arg\max_y \sum_{i=1}^M \rho_i \mathbb{1}[y = \arg\max_{y'} p_i(y'|x)] \ , \tag{2}$$

where $\mathbb{1}[\cdot]$ is 1 if its argument is true and zero otherwise. Uniform weighted aggregations are denoted by $\text{AVG}_u$ and $\text{MV}_u$.

### 2.1 Bayesian model average

In Bayesian inference, we update our prior belief about the parameters $w$ of a model that predicts $y$ given $x$ based on $p(y|x, w)$ to a posterior distribution $p(w|\mathcal{D})$ based on training data $\mathcal{D}$. Marginalizing $p(y|x, w)$ over this posterior gives the posterior predictive distribution

$$p(y|x, \mathcal{D}) = \int p(y|x, w)p(w|\mathcal{D})\mathrm{d}w, \tag{3}$$

also known as Bayesian Model Average (BMA), Bayes ensemble (e.g., Wenzel et al., 2020a), Bayesian marginalization (e.g., Wilson & Izmailov, 2020), and posterior predictive. It predicts by averaging all possible models weighted by their posterior probability. The BMA explicitly models aleatoric uncertainty (given by $p(y|x, w)$) and epistemic uncertainty in the form of $p(w|\mathcal{D})$ (see Caprio et al., 2024 for a recent discussion). As the integral in (3) is in general intractable, it is approximated in practice by an average of $M$ models sampled from the posterior:

$$p(y|x, \mathcal{D}) \approx \frac{1}{M} \sum_{m=1}^M p(y|x, w_m), \quad w_m \sim p(w|\mathcal{D}) \tag{4}$$

It is in general difficult to sample from $p(w|\mathcal{D})$, which can, for example, be addressed by Markov chain Monte Carlo (MCMC) sampling (Chen et al., 2014; Neal, 1992; 2011; Welling & Teh, 2011; Wenzel et al., 2020a; Zhang et al., 2020), Monte Carlo Dropout (Folgoc et al., 2021; Gal & Ghahramani, 2016), and Laplace approximation (MacKay, 1992b; Ritter et al., 2018).

### 2.2 Cancellation of errors

The cancellation of errors effect refers to the fact that, when individual ensemble members perform better than random guessing and make independent errors, their errors average out and the combined model outperforms the individual predictors (Eckhardt & Lee, 1985; Hansen & Salamon, 1990). Let us assume

binary classification and assume that each member of the ensemble has an error probability lower bounded by $p_{\max} < \frac{1}{2}$. Given $M$ models, the generalization error of the ensemble is upper bounded by

$$P_{(x,y)\sim\ p}(\mathrm{MV}_u(x) \neq y) \leq \exp\left(-2\big(\frac{M+1}{2} - Mp_{\max}\big)^2 M^{-1}\right)\ ,\tag{5}$$

see Appendix A.3 for a proof. That is, in the idealized (and not realistic) setting of independent errors, the generalization error vanishes with increasing $M$ if the classifiers are better than random guessing. The corresponding regression case relating the expected mean squared error and the correlation of the models' outputs is discussed in deep learning textbooks by (Goodfellow et al., 2016, sec. 7.11) and (Bishop & Bishop, 2023, sec. 9.6).

### 2.3 Deep ensembles

Deep ensembles (Lakshminarayanan et al., 2017) are ensembles with deep neural networks as members. More specifically, we refer to a deep ensemble if the networks all share the same structure. These networks are typically the result of independent training processes. The diversity is a result of random initialization and stochastic optimization. This randomness alone can yield well-performing ensembles, while additional randomization of the training data by bootstrap aggregation (bagging) bears the risk of the ensemble being worse than a single network trained on all data (Lakshminarayanan et al., 2017; Lee et al., 2015). Other ways to create deep ensemble members include the use of a cyclical learning rate schedule and taking the intermediate checkpoints as snapshot ensembles (SSEs) (Huang et al., 2017) as well as searching for other ensemble members in the neighborhood of a single pre-trained network (fast geometric ensembling (Garipov et al., 2018)). Although being introduced as an alternative to Bayesian approaches, Bayesian interpretations of deep ensembles can be found in recent work (D'Angelo & Fortuin, 2021; Wilson & Izmailov, 2020).

### 2.4 PAC-Bayesian majority voting

PAC-Bayesian analysis (McAllester, 1998; Valiant, 1984) provides bounds on the generalization error of Gibbs classifiers defined by a distribution $\rho$ over a (subset of a) hypothesis space $\mathcal{H}$ given empirical risks for the hypotheses.

A Gibbs classifier draws a hypothesis $h$ according to $\rho$ at random for each input $x$ and returns the prediction $h(x)$. PAC-Bayesian bounds hold for all distributions $\rho$ over $\mathcal{H}$ simultaneously. This allows us to directly optimize a PAC-Bayesian bound in terms of $\rho$, e.g., by gradient-based methods (Dziugaite & Roy, 2017; Masegosa et al., 2020; Masegosa, 2020; Ortega et al., 2022; Pérez-Ortiz et al., 2021; Thiemann et al., 2017). In a weighted majority voting classifier $\mathrm{MV}_\rho$ (e.g., a deep ensemble), the randomized prediction is replaced by a $\rho$-weighted vote by all hypotheses (Germain et al., 2009; 2015). PAC-Bayesian bounds can be applied to $\mathrm{MV}_\rho$ by bounding $L(\mathrm{MV}_\rho)$ by twice the risk of the corresponding Gibbs classifier (Germain et al., 2015; Langford & Shawe-Taylor, 2002; Masegosa et al., 2020). Gibbs classifiers ignore interactions between the models (such as cancellation of errors), and optimization of the weighting using bounds on the Gibbs classifier directly does typically not increase generalization performance of the ensemble (e.g., Lorenzen et al., 2019). This can be addressed by second-order PAC-Bayesian bounds (Germain et al., 2015; Lacasse et al., 2006; Masegosa et al., 2020). In particular, Masegosa et al. derive a bound in terms of the tandem loss

$$L(h, h') = \mathbb{E}_{p(x,y)}[\mathbb{1}(h(x) \neq y \wedge h'(x) \neq y)]\ ,\tag{6}$$

which takes pairwise correlations into account:

**Theorem 1** (Masegosa et al., 2020)**.** *For any probability distribution $\pi$ on $\mathcal{H}$ that is independent of $\mathcal{D}$ and any $\delta \in ]0, 1[$, with probability at least $1 - \delta$ over a random draw of $\mathcal{D}$ with $n$ elements, for all distributions $\rho$ on $\mathcal{H}$ and all $\lambda \in ]0, 2[$ simultaneously:*

$$L(\mathrm{MV}_\rho) \leq 4\left(\frac{\mathbb{E}_{\rho^2}[\hat{L}_\mathcal{D}(h, h')]}{1 - \lambda/2} + \frac{2\mathrm{KL}(\rho\|\pi) + \ln(2\sqrt{n}/\delta)}{\lambda(1 - \lambda/2)n}\right)\tag{7}$$

Here $KL(\rho\|\pi)$ denotes the Kullback–Leibler divergence between prior $\pi$ and posterior $\rho$, $\hat{L}_\mathcal{D}(h,h')$ is the empirical tandem loss on the data $\hat{L}_\mathcal{D}(h,h') = \frac{1}{|\mathcal{D}|}\sum_{(x,y)\in\mathcal{D}} \mathbb{1}(h(x) \neq y \wedge h'(x) \neq y)]$, and $\mathbb{E}_{\rho^2}$ is the short form for $\mathbb{E}_{h\sim\rho,h'\sim\rho}$. The bound can be efficiently optimized w.r.t. to $\rho$ and $\lambda$ (Thiemann et al., 2017), see Appendix A.4. In the following, we do not distinguish between the discrete distribution $\rho$ and the corresponding vector of probabilities $\rho \in \mathbb{R}^M$. The tandem loss is closely related to the disagreement-error ratio as defined by Theisen et al. (2023).[1]

We consider optimization of $\rho$ (i.e., the weights in a weighted ensemble) given $M$ models as proposed by Masegosa et al. (2020). Instead, one could also turn a second-order PAC-Bayesian bound into an objective function to optimize $\rho$ and the parameters of the $M$ models simultaneously. This was done by Masegosa (2020) for small multi-layer perceptrons and by Ortega et al. (2022) to optimize deep ensembles. However, because of the optimization of the neural networks' parameters and their weighting, the formal guarantees by the PAC-Bayesian bounds get lost. Furthermore, Ortega et al. updatet find improvements compared to uniformly weighted deep ensembles for larger networks (e.g., ResNet20, He et al., 2016) in their empirical evaluation.

## 3  Uniform, Bayesian, and PAC-Bayesian deep ensembles

In the following, we revisit results from the literature (Minka, 2002; Masegosa, 2020) and argue that general ensemble methods and BMA have different motivations, which can lead to differences in prediction performance in practice. Bayes ensembles are a way to improve uncertainty quantification and calibration of DNNs and often increase accuracy compared to a single neural network. However, the BMA does not take interactions between the ensemble members into account and weights them according to $p(w|\mathcal{D})$, which does not necessarily align with maximizing the predictive performance of an ensemble.

### 3.1  Bayes ensembles and cancellation of errors

We first discuss the limit behavior of the BMA Bayes ensemble as defined in (4) for increasing training data set size. Then we will discuss this result in the context of ensembles of deep neural networks. After that, we bring forward the approach of using validation data for PAC-Bayesian weighting instead of using the data for early-stopping.

#### 3.1.1  Limit behavior of the BMA

In a weighted additive ensemble, a finite set of hypotheses from $\mathcal{H}$ are combined to a new model $h_\rho \in \mathcal{H}_\mathrm{E}$ from the hypothesis class

$$\mathcal{H}_\mathrm{E} = \left\{ h_\rho(x|h_1,\ldots,h_M) \,\middle|\, M \in \mathbb{N}, \rho \in \mathbb{R}^M, h_i \in \mathcal{H} \right\}. \tag{8}$$

Learning a weighted combination of hypotheses can be regarded as a form of stacked generalization (Wolpert, 1992). As discussed in Section 2.2, $h_\rho$ can generalize better than the individual $h_1,\ldots,h_M$ if the ensemble members are diverse in the sense that their errors are not strongly correlated. Typically, $\mathcal{H} \subset \mathcal{H}_\mathrm{E}$, and for most ensemble methods this is key. The ensemble members selected from $\mathcal{H}$ are often referred to as weak learners and are not expected to have a very low risk (generalization performance) as long as they are better than random guessing: The desired model $h_\mathrm{E}^*$ is expected to be in $\mathcal{H}_\mathrm{E} \setminus \mathcal{H}$. Boosting with decision stumps is a prime example (Freund et al., 1996).

In Bayesian machine learning, the posterior distribution $p(h|\mathcal{D})$ over $\mathcal{H}$ models the uncertainty of identifying a desired model $h^* \in \mathcal{H}$ given finite training data $\mathcal{D}$ and the prior over $\mathcal{H}$.[2] In Bayesian deep learning with a fixed neural network architecture $h_w$ parameterized by weights $w \in \mathbb{R}^d$, we have $\mathcal{H} = \{h_w | w \in \mathbb{R}^d\}$ and the posterior over the hypotheses is modelled by a distribution $p(w \,|\, \mathcal{D})$ over the weights $w$. To simplify

---

[1]The theoretical results by Theisen et al. (2023) are derived under an additional assumption on the performance of the ensemble members, which we avoid here.

[2]The desired model in $\mathcal{H}$ is the model in $\mathcal{H}$ closest to the true model $h_p$ underlying the data generating distribution $p$. We do neither assume $h_p \in \mathcal{H}$ nor $h_p \in \mathcal{H}_\mathrm{E}$.

the discussion, let us assume that the models are parameterized in a way that ensures identifiability, that is, different parameters generate different hypotheses. With increasing training data set size $n = |\mathcal{D}|$ the uncertainty about the desired model decreases, and $p(w\,|\,\mathcal{D})$ should be concentrated around $w^*$ (i.e., the weights of the desired model $h^* = h_{w^*}$). This is formally shown by the Bernstein-von Mises theorem (Kleijn & Van der Vaart, 2012; Van der Vaart, 2000), which was first discovered by Laplace (1820), taken up by Bernstein (1917) and Von Mises (1931) independently, and proven in a more general form by Le Cam (1953), who coined the name.

**Theorem 2** (Bernstein-von Mises theorem). *Let the parametric model $h_w$ be twice differentiable with a non-singular Fisher information matrix $\mathcal{I}(w^*)$ at $w^*$, identifiable and continuous on a compact parameter space. Furthermore, let the prior measure be absolutely continuous in a neighbourhood of $w^*$ with a continuous density non-zero at $w^*$. Let us consider the sequence of posterior distributions $p_n(w\,|\,\mathcal{D}_n)$ and maximum likelihood estimates $\hat{w}_n$ given $\mathcal{D}_n$ for data sets $\mathcal{D}_n \subset \mathcal{D}_{n+1}$. The Bernstein-von Mises theorem implies*

$$\|p_n(w|\mathcal{D}_n) - \mathcal{N}(\hat{w}_n, n^{-1}\mathcal{I}^{-1}(w^*))\|_{TV} \overset{p_{w^*}}{\to} 0. \tag{9}$$

That is, for increasing data set size, the posterior converges in probability to a normal distribution $\mathcal{N}(\hat{w}_n, n^{-1}\mathcal{I}^{-1}(w^*))$, where $\|\cdot\|_{\mathrm{TV}}$ denotes the total variation norm (see Van der Vaart, 2000). The normal distribution is centered on the maximum likelihood estimate with covariance $n^{-1}\mathcal{I}^{-1}(w^*)$, that is, the distribution gets more and more concentrated with increasing data set size. Under the same assumptions, this implies that the expectation of the Bayes ensemble (4) converges to the maximum likelihood estimate $h_{w^*} \in \mathcal{H}$ for $n \to \infty$ (while the Bayes ensemble is not necessarily in $\mathcal{H}$ for finite $n$).

Thus, as already argued by Masegosa (2020, sec. 4) and Minka (2002), from the perspective of general ensemble learning and the goal of finding a model with minimum risk, a sample from the posterior $p(w\,|\,\mathcal{D})$ does not appear to be a particularly good choice for ensemble members. As the sample will be distributed around the maximum likelihood estimate $h_{w^*} \in \mathcal{H}$, it cannot be expected that the members of the ensemble will be very diverse, and the lack of diversity reduces the cancellation of errors effect. The Bayes ensemble will tend to the model $h^*$ with the highest likelihood in $\mathcal{H}$ – and not to the model $h_{\mathrm{E}}^*$ with the highest likelihood in $\mathcal{H}_{\mathrm{E}}$. More formally:

**Corollary 1.** *Under the conditions of Theorem 2, if $\min_{h\in\mathcal{H}} L(h) > \min_{h\in\mathcal{H}_E} L(h)$, the Bayes ensemble as defined in* (4) *will not converge to the best generalizing ensemble solution for increasing training data set size.*

*Proof.* This follows directly from Theorem 2, which implies that the BMA converges to the maximum likelihood solution corresponding to $h^* = \arg\min_{h\in\mathcal{H}} L(h)$, which is by assumption worse than the optimal ensemble $h_{\mathrm{E}}^* = \arg\min_{h\in\mathcal{H}_{\mathrm{E}}} L(h)$. □

### 3.1.2 BMA-based deep neural network ensembles

Corollary 1 leads to the following:

**Claim 1.** *The Bayes ensemble is not a particularly good way to select and weight networks in a deep ensemble if the goal is to create the best generalizing ensemble.*

Although this claim appears to be evident given the theoretical considerations above (see also, e.g., Minka, 2002; Masegosa, 2020), many studies evaluate the performance of the Bayes ensemble by measuring generalization performance on a test set and provide favorable comparisons with baselines, especially single networks, suggesting the posterior predictive as a way to maximize generalization performance.

One may well argue that the asymptotic argument in Corollary 1 is of little practical relevance if $\mathcal{H}$ already has a high capacity (e.g., consists of overparameterized deep neural networks), because then $h_{w^*} \in \mathcal{H}$ may be close to the optimal solution, and even for sufficiently large finite $n$ the $M$ ensemble members can be expected to be quite similar in their predictive behavior. If ensemble members become so similar that there is no cancellation of errors effect, the ensemble will not improve over the performance of a single ensemble member. This was studied experimentally by Kobayashi et al. (2021) for increasing complexity of the ensemble

members. Thus, to test Claim 1 for finite (and not very large) training set sizes, we conducted empirical comparisons of Bayes ensembles with alternative ensemble approaches, see section 4.

A Bayesian way to find the best generalizing ensemble is to lift the Bayesian reasoning from $\mathcal{H}$ to $\mathcal{H}_{\mathrm{E}}$. This has been demonstrated by Monteith et al. (2011), who apply BMA not to $h_w \in \mathcal{H}$, but to $h_\rho \in \mathcal{H}_{\mathrm{E}}$, where $\mathcal{H}_{\mathrm{E}}$ are all ensembles of a constant size $M$. In such an approach, the posterior distribution contracts around the maximum likelihood estimate of the best weighting of ensemble members. Thus, we go from a distribution over the weights of a single neural network to a distribution over all parameters of the $M$ networks and the ensemble weights. This growth in the number of dimensions renders the approach computationally infeasible in larger settings.

The most basic way to create an ensemble of $M$ neural networks is to repeat the training process $M$ times using all training data and to combine the resulting networks uniformly (Hansen & Salamon, 1990; Lakshminarayanan et al., 2017). This procedure can be viewed as a simple ensembling, where random weight initializations and stochastic training (randomly composed mini-batches, randomized augmentations), in general stopped before convergence, are assumed to generate sufficiently diverse ensemble members. Although this approach is often explicitly referred to as non-Bayesian (see, e.g., Lakshminarayanan et al., 2017; Ovadia et al., 2019; Wenzel et al., 2020a), the resulting ensemble may also be interpreted as a BMA from an approximate posterior (Gustafsson et al., 2020; Wilson & Izmailov, 2020). In the former case, the motivation is to generate a diverse set of ensemble members (i.e., to find a good solution in $\mathcal{H}_{\mathrm{E}}$), in the latter the focus is on the uncertainty about the single best model (in $\mathcal{H}$). When modifying the algorithm, for example, by introducing non-uniform weighting of the ensemble members, the difference in motivation matters. For example, *putting little weight on the hypothesis with maximum likelihood can be reasonable from the ensembling point of view, but is difficult to justify from the BMA perspective.*

In BMA it is also acknowledged that having samples with the same or very similar behavior in (4) is not very efficient and methods have been devised to approximate the BMA using a diverse set of samples (Wilson & Izmailov, 2020), but "[u]ltimately, the goal [of BMA] is to accurately compute the predictive distribution" in (3) (Wilson & Izmailov, 2020), which is a different goal than finding the hypothesis with the lowest risk in $\mathcal{H}_{\mathrm{E}}$ (Minka, 2002). *At the core of the Bayesian ensembling approach is the weighting based on $p(w|\mathcal{D})$ – and this is suboptimal because it ignores the interactions between ensemble members.*[3]

### 3.2 PAC-Bayesian deep ensembles

Claim 1 does not state that one cannot do better than uniform averaging given $M$ models from $\mathcal{H}$ (as already pointed out in early works on ensembles, e.g., Krogh & Sollich, 1997) nor that we cannot make use of priors and Bayesian reasoning to improve ensembles. For example, there is no need to put weight on an ensemble member that is worse than random guessing or always errs when another ensemble member errs (i.e., it can never correct an error).

Given the results by Masegosa et al. (2020) and Wu et al. (2021), it appears to be promising to weight the members of the ensemble based on PAC-Bayesian generalization bounds optimized using the tandem loss. It is crucial to consider a second-order bound that takes interactions between ensemble members – error cancellation – into account; otherwise the approach suffers from the same problem as BMA and concentrates all weight on a single ensemble member in the limit.

The distribution $\rho$ found by optimization based on Theorem 1, which weights the ensemble members, is also a posterior distribution, but different from $p(w \,|\, \mathcal{D})$ in the BMA. The distribution $\rho$ optimized using the tandem loss is qualitatively different from $p(w|\mathcal{D})$, because it can be viewed as being optimized to improve performance (of a surrogate loss) over $\mathcal{H} \times \mathcal{H}$ (see the $\mathbb{E}_{h \sim \rho, h' \sim \rho}$ term in Theorem 1). One can view $\rho$ as parameterizing distributions over $\mathcal{H}^2$, sitting between $p(w|\mathcal{D})$ over $\mathcal{H}$ and to modelling a full distribution over $\mathcal{H}^M$ (e.g., Monteith et al., 2011).

The second-order bounds require for each ensemble member $h_i$ that there are data points $\mathcal{D}_i$ not used for training $h_i$ and that for each pair of models $h_i$ and $h_j$ the overlap $\mathcal{D}_i \cap \mathcal{D}_j$ is large enough for a good enough estimate of the tandem loss: In (7), we compute $\hat{L}_{\mathcal{D}_i \cap \mathcal{D}_j}(h_i, h_j)$ for each pair of hypotheses $h_i$ and $h_j$ and $n$

---

[3]The original BMA (4) is based on independent draws from $p(w|\mathcal{D})$, thus the $w_1, \ldots, w_M$ are independent by definition.

becomes $n = \min_i |\mathcal{D}_i|$. For random forests as considered by Lorenzen et al. (2019) and Masegosa et al. (2020), data sets $\mathcal{D}_i \subset \mathcal{D}$ naturally arise from the bagging procedure and the proposed bounds can be computed and optimized for free. When applied to deep ensembles, we have to pay by leaving data out when training the ensemble members, which can be expected to reduce the performance of the individual networks. Taking this into account, we claim:

**Claim 2.** *PAC-Bayesian weighting optimized using the tandem loss can improve the generalization performance of a deep ensemble, in particular if extra data for computing the bound are available, and provide nonvacuous performance guarantees.*

Theorem 1 based on the tandem loss provides a bound on the classification error that holds simultaneously for all distributions $\rho$. Thus, given hypotheses $h_{w_1}, \ldots, h_{w_M}$ and data sets $\mathcal{D}_i$, we suggest optimizing the second-order PAC-Bayesian bound to adapt $\rho$, which can be done very efficiently using gradient-based optimization, see Appendix A.4 for details about the optimization process following Wu et al. (2021).

Because Theorem 1 holds for any $\rho$, it provides rigorous performance guarantees even if $\rho$ is optimized using the data $\mathcal{D}_i$. *That is the fundamental difference to the work by Masegosa (2020) and Ortega et al. (2022)*, who derive a loss function based on a PAC-Bayesian bound and optimize the parameters of the hypotheses $w_1, \ldots, w_M$. In this approach, the optimized loss functions do not provide performance guarantees, because the models $h_{w_1}, \ldots, h_{w_M}$ are not independent of the data used for computing the loss (the same would happen if we would optimize both $\rho$ and $w_1, \ldots, w_M$ based on Theorem 1. Masegosa (2020) proves a PAC-Bayesian bound on the cross-entropy that takes correlations into account. A loss function is derived on the basis of this bound, which is then optimized to build ensembles. The study considers ensembles of multi-layer perceptrons with 20 hidden units, which are jointly trained by minimizing the new loss. Ortega et al. (2022) extend this work, which is, to our knowledge, the only study applying PAC-Bayesian methods to DNN ensembles. They empirically compare uniform DNN ensembles with ensembles of DNNs that are jointly trained with newly proposed loss functions derived from a PAC-Bayesian bound on the cross-entropy. For larger DNNs such as ResNet20, the authors find that direct optimization of their bound does not induce ensembles with much higher diversity and with similar or even worse performance than uniform deep ensembles with individually trained members, which is in contrast to our empirical results described in section 4. Apart from unsatisfactory empirical results when applied to DNNs, the bounds on the cross-entropy underlying the algorithms in Masegosa (2020) and Ortega et al. (2022) are not valid anymore after constructing the ensemble. This is in contrast to the approach proposed here, using the 0-1 loss bound given in Theorem 1 to weight trained models. The bound holds for all $\lambda$ and $\rho$ simultaneously, and we optimize $\lambda$ and $\rho$ on independent data, which can be done efficiently as described in Appendix A.4.

### 3.2.1 PAC-Bayesian weighing and early-stopping

In most deep ensemble approaches, each ensemble member is the result of an independent training process, where the weight configuration is chosen by early-stopping (requiring some hold-out data) or simply after a predefined number of learning iterations (which is then also a crucial hyperparameter). In contrast, Wenzel et al. (2020a) sample from the posterior using a single stochastic gradient Markov chain Monte Carlo (SG-MCMC) sequence and SSEs combine networks from a single learning process (Huang et al., 2017). That is, one process creates all members of the ensemble. Having a theoretically sound way to weight models in an ensemble reduces the risk that some of the models deteriorate the overall performance. The second-order PAC-Bayesian weighting performs model selection taking both individual performance and diversity into account, which should provide protection from networks that do not contribute positively to an ensemble. It allows one to add models to the ensemble that do not perform particularly well and/or are correlated. As an example, this makes it safe to include several weight configurations (checkpoints) from a single learning process to an ensemble. It also renders using a hold-out validation dataset for early-stopping the neural network training unnecessary: It should neither be harmful to add underfitted nor overfitted models. The early-stopping data can instead be used to optimize the PAC-Bayesian weighting:

**Claim 3.** *Optimizing the weighting using the tandem loss allows inclusion of several models from a training run in a way that can efficiently improve performance and make early-stopping unnecessary.*

In the following, we will study the three claims empirically.

## 4 Experiments and results

### 4.1 Experimental setup

Four different datasets and four neural network architectures were considered.[4] We evaluated a CNN LSTM (Yenter & Verma, 2017) on the IMDB binary classification benchmark (Maas et al., 2011) and compared it with a state-of-the-art Bayesian approximate inference method from Wenzel et al. (2020a) (referred to as cSGHMC-ap), who combine stochastic gradient Langevin dynamics (SGLD) (Chen et al., 2014; Welling & Teh, 2011) with a cyclical learning rate schedule (Zhang et al., 2020), and adaptive preconditioning (Li et al., 2016; Ma et al., 2015). Furthermore, we considered the multi-class data sets CIFAR-10 and CIFAR-100 (Krizhevsky, 2009) with ResNet110 (He et al., 2016) and WideResNet28-10 (WRN28-10) (Zagoruyko & Komodakis, 2016) architectures, for which Ashukha et al. (2020) and Vadera et al. (2022) provide reference results for a variety of Bayesian methods. Lastly, the ResNet50 was evaluated on the EyePACS dataset that contains diabetic retinopathy diagnoses distinguishing five degrees of severity. It was introduced by Band et al. (2021) as a benchmark for Bayesian approximate inference methods. The authors provide results for a variety of Bayesian neural networks and ensembles of those, creating "deep ensembles of Bayes ensembles". As we are concerned with a comparison to Bayes ensembles, we focus on the results of the individual Bayesian methods as a reference. The hyperparameters for each experiment are given in Appendix A.4.

For each dataset and model architecture, a simple deep ensemble was constructed based on the reference paper's single-run hyperparameters. Intermediate checkpoints were stored and either included (referred to as *all* from now) or ignored in the final ensemble (*last*). Further, ensembles with sub-sampled training sets (bagging) as well as snapshot ensembles (SSE) using a cyclical learning rate were formed. Finally, the trade-off between the size of the ensemble and the number of training epochs was studied in a sequential setting in Appendix A.5.2.

We trained $M_{\text{total}}$ networks for each setting, where $M_{\text{total}}$ was 30, 50, and 10 for CIFAR, IMDB, and EyePACS, respectively. Ensembles of size $M$ were created by sampling without replacement from these models. This was repeated five times for every setting, and we report mean and standard deviation $\sigma$ of these five trials. The optimization of the tandem loss followed Wu et al. (2021), see Appendix A.4 for details. Generalization bounds are reported for $\delta = 0.05$.

**Test-time cross validation.** To optimize the weighting while retaining the original training data set size, *test-time cross validation* as introduced by Ashukha et al. (2020) was employed. We used 50% of the hold-out data for bound optimization and the other half for testing. This was then repeated with the two subsets switching roles, and the results were averaged to decrease the variance for the unbiased performance estimate, see Appendix A.2 for details.

**IMDB validation data.** We took the data sets and models from the BMA literature. The standard training protocol for IMDB involves a validation data set to avoid overfitting. The neural networks are trained on the training data set, and the weight configuration during training that achieves the lowest error on the validation data set is considered the result of the training process.

For the other data sets and models considered in this study, this type of early-stopping is not necessary to avoid overfitting, therefore the data is used additionally for training. Thus, these settings are actually not well suited to show the benefits of using PAC-Bayesian weighting as an approach to sort out underperforming checkpoints.

The IMDB setting allows us to compare uniform deep ensembling using early-stopping with PAC-Bayesian ensembling without early-stopping but using the validation data for optimizing the PAC-Bayesian bound. In this setting, there is no potential bias introduced by test-time cross validation due to different amounts of data used in the algorithms.

**Rerunning baseline experiments.** The experiments from Wenzel et al. (2020a) on IMDB and CIFAR-10 ResNet20 were repeated with the source code provided by the authors to obtain exact accuracies. The results

---

[4]We consumed approximately 320 GPU days for all experiments on a local cluster.

on our compute system deviated from those reported by Wenzel et al. (2020a). On IMDB, the best accuracy we achieved was 0.860 with $T = 0.562$ (the temperature parameter of the temperized posterior) compared to $\approx 0.870$ (interpolated) in the plot in the original publication. For the ResNet20 on CIFAR-10, the best accuracy we achieved was 0.917 with $T = 0.0316$ compared to $\approx 0.936$ in the plot the original publication.

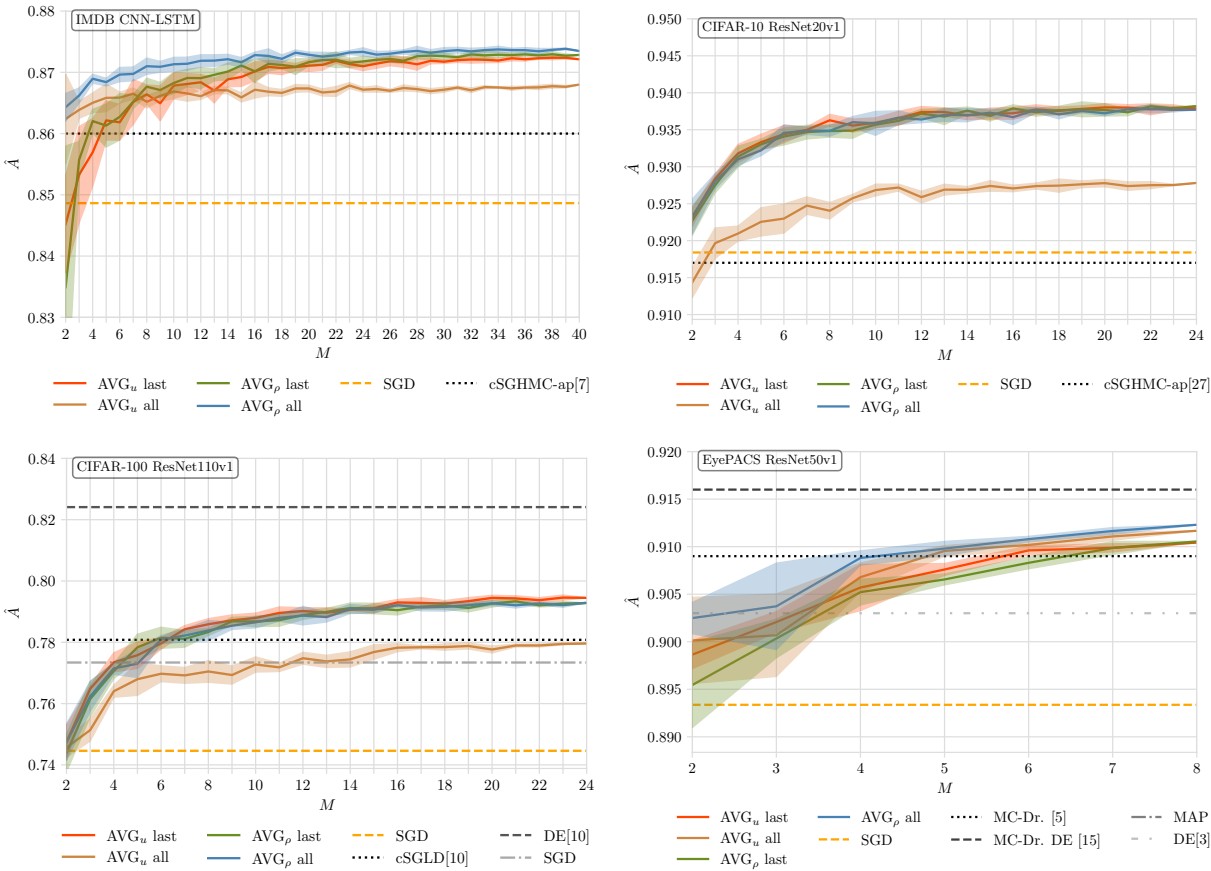

Figure 1: Mean test accuracy $\hat{A} \pm \sigma$ vs. ensemble size $M$ over five ensembles for uniformly and PAC-Bayesian weighted deep ensembles ($\text{AVG}_u$ and $\text{AVG}_\rho$), using either only the last or all training checkpoints, and the best single model (SGD). References are Bayesian ensembles cSGHMC-ap (Wenzel et al., 2020a), cSGLD (Ashukha et al., 2020), *MC-Dr*opout and *MC-Dr*opout *D*eep *E*nsemble (i.e., ensemble of Bayesian ensembles), as well as simple *D*eep *E*nsembles for EyePACS from Band et al. (2021). Numbers in brackets indicate the ensemble sizes of the baselines, the ensemble sizes are taken from the literature. Additional results for other settings are shown in Figure A.5 in the appendix.

## 4.2 Results

**Bayesian vs. uniform deep ensembles (Claim 1).** We compared the uniformly weighted deep ensembles across all datasets and model architectures with Bayesian ensembles from the literature. Our results shown in Table 1 and figures 1 and A.5 demonstrate that uniformly weighted deep ensembles can match the performance of models based on BMA. Looking at Table 1, we see that a basic uniform deep ensemble ($\hat{A}(\text{AVG}_u)$, *last*, *Simple*) always performed at least as well as the BMA baselines.

The ensembles in Table 1 have different sizes; the ensemble sizes for the baselines were taken from the literature. The ensemble sizes for the uniform and PAC-Bayesian ensembles are varied in figures 1 and A.5. On IMDB, a uniform deep ensemble of size $M = 7$ improved on our BMA results using the code from Wenzel et al. (2020a). On CIFAR-10 and ResNet20, the uniform ensemble already outperformed the Wenzel et al. baseline for fewer than 27 ensemble members.

Table 1: Mean test accuracies $\hat{A}$ over five ensembles, ensemble size is given in brackets (for other ensemble sizes see figures A.5, A.6, A.7, and A.8 in the appendix). The *Simple* results refer to simple deep ensembles with SGD hyperparameters from the reference papers. The attributes *last* and *all* indicate whether only the last (a single) or all training checkpoints of a training run are included in the ensemble. *Bayesian reference* results are taken from Ashukha et al. (2020) (CIFAR ResNet110 & WRN28-10), Band et al. (2021) (EyePACS) and our reproduction of the experiments by Wenzel et al. (2020a) (IMDB, CIFAR-10 ResNet20).

| Model (Dataset) | Experiment | $\hat{A}(\mathrm{AVG}_u)$ last | $\hat{A}(\mathrm{AVG}_u)$ all | SGD | Bayesian reference |
|---|---|---|---|---|---|
| CNN LSTM (IMDB) | Simple | 0.871[40] | – | 0.853 | |
| | Checkp. | **0.872**[40] | 0.868[40·5] | 0.849 | 0.860[7] |
| | SSE | 0.729[8] | 0.858[8·10] | 0.861 | |
| ResNet20 (CIFAR-10) | Simple | **0.938[24]** | 0.928[24·5] | 0.918 | 0.917[27] |
| | SSE | 0.894[24] | 0.903[24·5] | 0.896 | |
| ResNet110 (CIFAR-10) | Simple | **0.956[20]** | 0.944[20·5] | 0.943 | 0.955[10] |
| | SSE | 0.954[20] | 0.951[20·5] | 0.941 | |
| WRN28-10 (CIFAR-10) | Simple | **0.967[24]** | 0.96[24·5] | 0.961 | **0.967[10]** |
| | SSE | 0.964[24] | 0.964[24·5] | 0.956 | |
| ResNet110 (CIFAR-100) | Simple | **0.794[24]** | 0.78[24·5] | 0.745 | 0.781[10] |
| | SSE | 0.79[24] | 0.788[24·5] | 0.733 | |
| WRN28-10 (CIFAR-100) | Simple | **0.83[24]** | 0.827[24·5] | 0.798 | 0.828[10] |
| | SSE | 0.818[24] | 0.822[24·5] | 0.791 | |
| ResNet50 (EyePACS) | Simple | 0.910[8] | **0.912**[8·6] | 0.895 | 0.909[5] |

On CIFAR-100 with a ResNet110, the uniform deep ensemble outperformed the Bayesian reference from Ashukha et al. (2020) by 1.3 ppt. Interestingly, the SGD and deep ensemble results from Ashukha et al. are notably better than ours. As the authors base their Bayesian methods on pre-trained SGD solutions, we can assume that their Bayesian reference performances could also profit from the better accuracies achieved in their training environment. Thus, it is remarkable that our uniform deep ensembles nevertheless matched the Bayesian references, and we would expect an even better performance if compared in the same environment.

In accordance with the literature, bagging reduced the performance, see Appendix A.5.1 for details. The experiments on the trade-off between $M$ and the number of training epochs showed that, even in a sequential setting with a strict limit on the number of overall training epochs, deep ensembles could match the BMA approach, see Appendix A.5.2 for details.

**Bayesian vs. PAC-Bayesian deep ensembles (Claim 2).** The results of our comparison of Bayesian and PAC-Bayesian deep ensembles are summarized in Table 2, Figure 2, Table 3 , and also figures 1 and A.5.

First, let us look at the results using test-time cross validation in Table 2. Optimizing the weighting by minimizing the second-order PAC-Bayesian bound matched the uniform performance in all cases $\pm 0.1$ ppt, while even slightly improving the ensemble performance on IMDB (Table 2: $\hat{A}(\mathrm{AVG}_u)$ *last* vs. $\hat{A}(\mathrm{AVG}_\rho)$ *last*). At the same time, the PAC generalization guarantees in Table 3 – which still hold after optimization – tightened dramatically, most significantly on CIFAR-10 ResNet20 by 31.8 ppt, from 0.425 to 0.743. For ResNet110 on CIFAR-100, the optimization was necessary to obtain a non-trivial bound, increasing from 0.0 to 0.127. The PAC-Bayesian guarantees hold for $\mathrm{MV}_\rho$ as the aggregation method, and $\mathrm{AVG}_\rho$ usually performs slightly better in practice. However, Table A.5 in the appendix shows how similar the two aggregation methods behave. That is, a slight decrease in performance can give rigorous generalization bounds.

Table 2: Mean test accuracies $\hat{A}$ over five ensembles, ensemble size is given in brackets (for other ensemble sizes see figures A.5, A.6, A.7, and A.8 in the appendix). The *Simple* results refer to simple deep ensembles with SGD hyperparameters from the reference papers. The attributes *last* and *all* indicate whether only the last (a single) or all training checkpoints of a training run are included in the ensemble. The subscript $\rho$ indicates that the weighting of the ensemble members was based on minimizing a PAC-Bayesian tandem bound using test-time cross validation.

| Model (Dataset) | Experiment | $\hat{A}(\text{AVG}_u)$ last | $\hat{A}(\text{AVG}_\rho)$ last | $\hat{A}(\text{AVG}_u)$ all | $\hat{A}(\text{AVG}_\rho)$ all |
|---|---|---|---|---|---|
| CNN LSTM (IMDB) | Simple | 0.871[40] | 0.872[40] | – | – |
| | Checkp. | 0.872[40] | 0.873[40] | 0.868[40·5] | 0.873[40·5] |
| | SSE | 0.729[8] | 0.741[8] | 0.858[8·10] | **0.874[8·10]** |
| ResNet20 (CIFAR-10) | Simple | **0.938[24]** | **0.938[24]** | 0.928[24·5] | **0.938[24·5]** |
| | SSE | 0.894[24] | 0.894[24] | 0.903[24·5] | 0.908[24·5] |
| ResNet110 (CIFAR-10) | Simple | **0.956[20]** | **0.956[20]** | 0.944[20·5] | **0.956[20·5]** |
| | SSE | 0.954[20] | 0.955[20] | 0.951[20·5] | 0.954[20·5] |
| WRN28-10 (CIFAR-10) | Simple | **0.967[24]** | **0.967[24]** | 0.96[24·5] | **0.967[24·5]** |
| | SSE | 0.964[24] | 0.963[24] | 0.964[24·5] | 0.964[24·5] |
| ResNet110 (CIFAR-100) | Simple | **0.794[24]** | 0.793[24] | 0.78[24·5] | 0.793[24·5] |
| | SSE | 0.79[24] | 0.79[24] | 0.788[24·5] | **0.794[24·5]** |
| WRN28-10 (CIFAR-100) | Simple | **0.83[24]** | **0.83[24]** | 0.827[24·5] | 0.829[24·5] |
| | SSE | 0.818[24] | 0.818[24] | 0.822[24·5] | 0.826[24·5] |
| ResNet50 (EyePACS) | Simple | 0.91[8] | 0.91[8] | 0.912[8·6] | **0.913[8·6]** |

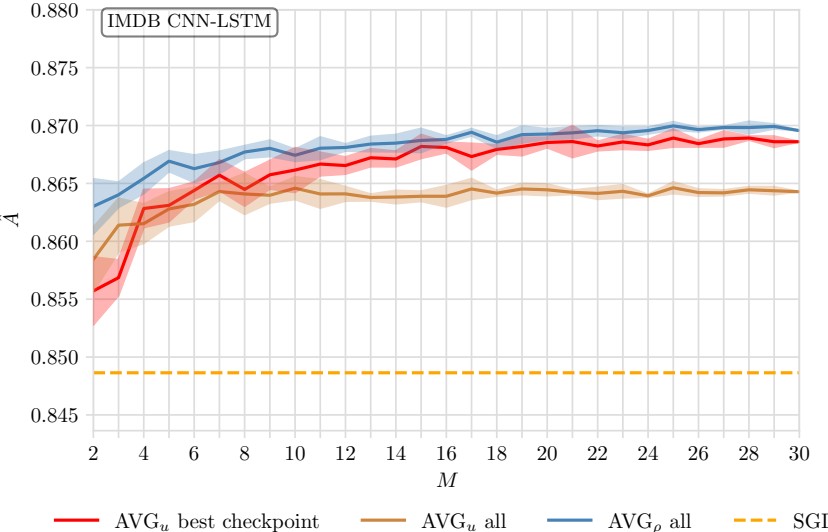

Figure 2: Mean test accuracy $\hat{A} \pm \sigma$ vs. ensemble size $M$ over five ensembles for uniformly and PAC-Bayesian weighted deep ensembles, using either only a single or all training checkpoints, where the validation data was used to either select the best checkpoint ($\mathrm{AVG}_u$ *best checkpoint*) or optimize the weights ($\mathrm{AVG}_\rho$ *all*).

Figures 3 and 4 exemplify the weighting from minimizing the tandem loss and a first-order PAC generalization bound. They show that the second-order objective function avoids putting all weight on a single hypothesis, while the first-order bound optimization does not.

Computing and optimizing the PAC-Bayesian bound requires additional hold-out data, which biases the comparison with uniform weighting. Instead of using additional data, one could use bagging (Lorenzen et al., 2019; Masegosa et al., 2020). The results of our bagging experiments are presented in Appendix A.5.1. For our ensembles of neural networks and rather small data sets, bagging decreased the performance, which is in line with the literature (Lakshminarayanan et al., 2017; Lee et al., 2015).

Figure 2 depicts the result when a validation set can be used to optimize the PAC-Bayesian bound. Here, the uniform deep ensemble used a validation data set to pick the weight configuration from each training run ($\mathrm{AVG}_u$ *best checkpoint*), which performed better than uniformly averaging over all checkpoints ($\mathrm{AVG}_u$ *all*). The PAC-Bayesian approach used the validation data to optimize the bound instead, that is, all algorithms have exactly the same data available for constructing the ensemble. Figure 2 shows that the PAC-Bayesian ensembles optimizing $\rho$ over all snapshots ($\mathrm{AVG}_\rho$ *all*) outperformed the uniform ensemble using the validation data to select snapshots ($\mathrm{AVG}_u$ *best checkpoint*).

**Bayesian vs. second-order PAC-Bayesian deep snapshot ensembles (Claim 3).** Let us again start with the results using test-time cross-validation in Table 2. We evaluated snapshot ensembling (Huang et al., 2017) as well as simply taking checkpoints from the original learning rate schedule. In SSEs, going from one to all intermediate checkpoints with optimized weighting improved predictive performance in all but one case. On IMDB, this improvement was most pronounced with 13.3ppt (Table 2: $\hat{A}(\mathrm{AVG}_\rho)$ *last* vs. $\hat{A}(\mathrm{AVG}_\rho)$ *all*). An SSE with 8 members (and 10 snapshots per member) gave the best accuracy with a training budget ($8 \cdot 50 = 400$ epochs) below the budget of the Bayesian reference method (500 epochs). SSEs include networks when the learning rate is lowest, just before re-starting the learning rate cycle. However, including checkpoints following the original schedule also matched the baseline of taking only the last checkpoint across all experiments. This straightforward approach improved performance on IMDB and EyePACS (Table 2: $\hat{A}(\mathrm{AVG}_\rho)$ *last* vs. $\hat{A}(\mathrm{AVG}_\rho)$ *all*). This is in contrast to the uniform weighting results, where adding the snapshots decreased the performance compared to only taking the last network.

These results do not change when looking at the results for the experiments where a validation set was available shown in Figure 2. Here, using all snapshots decreased the performance of the uniform ensembles, $\hat{A}(\text{AVG}_u)$ *last* vs. $\hat{A}(\text{AVG}_u)$ *all*, while the PAC-Bayesian $\hat{A}(\text{AVG}_\rho)$ *all* performed best.

As hypothesized, the tandem loss bound optimization allows one to include worse performing models in the ensemble (at no additional training cost) while maintaining or improving performance. Figures 3 and 4 show that the second-order PAC-Bayesian approach picks snapshots from the different training runs. This can be seen most clearly from the regular pattern in Figure 4, bottom left. Figure 4 shows the differences between the *Simple* and *SSE* training setup. In the former, there was a tendency to pick the last snapshots of each network training run; in the latter, which used a cyclic learning rate schedule, intermediate snapshots were also selected.

Thus, optimizing the weights is crucial when considering all models from a training run. We attribute this to the importance of filtering out models with low performance. When considering all models and optimizing the weighting, the *Simple* setting using SGD hyperparameters from the reference papers performed on par with the SSE setting using a cyclic learning rate. Thus, using a special learning rate schedule when combining models from a single training run was not necessary when the proposed weighting method was employed.

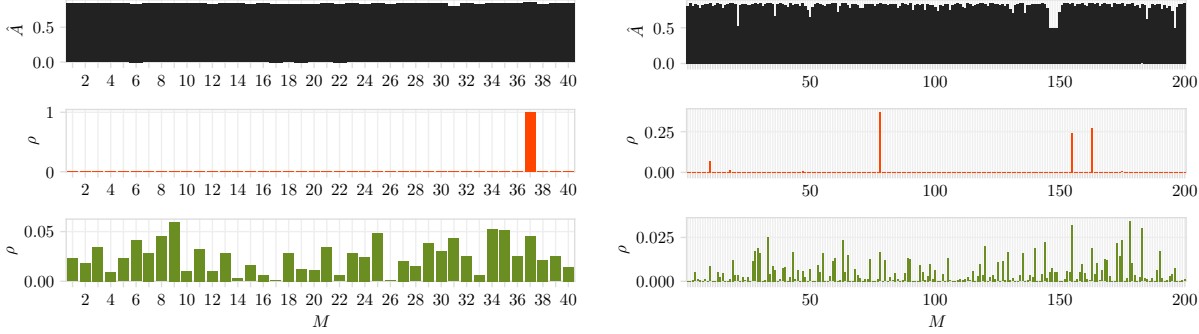

Figure 3: IMDB accuracies ($\hat{A}$, top) and weight distribution per member for first-order (middle, Lorenzen et al., 2019) and tandem bound weighting (bottom) for the *Simple* setting considering only a single network per training process (left) and when adding checkpoints (right). The number of training runs was 40. Four intermediate checkpoints were added, giving a total of five weight configurations per training process.

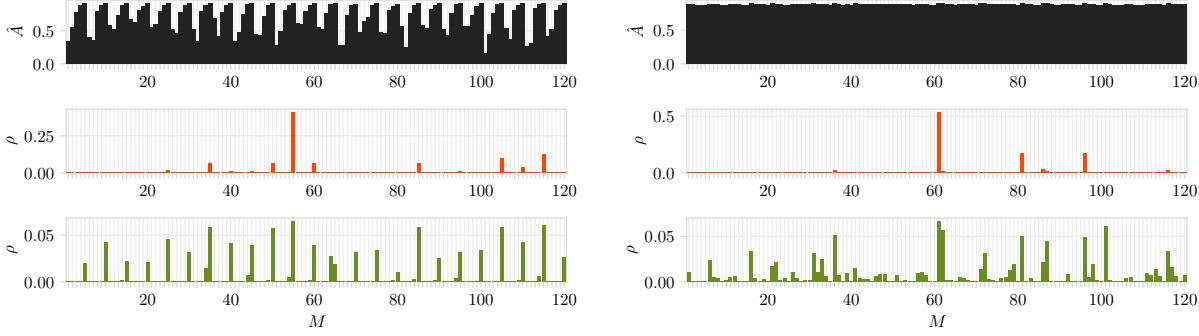

Figure 4: ResNet20 on CIFAR-10 accuracies (top) and weight distribution per member for first-order (middle) and tandem bound weighting (bottom) for the *Simple* setting (left) and *SSE* (right), both including all checkpoints (24 training processes, five checkpoints per process).

Table 3: Test accuracies for $\text{AVG}_\rho$ and $\text{MV}_\rho$ and PAC-Bayesian bounds ($\delta = 0.05$) for all *Simple* experiments (see Table 2). An overview over all experiments is shown in Figure A.5 in the appendix. The bounds were computed for the *all* setting, except for IMDB, where checkpoints were taken in a separate experiment due to early-stopping in *Simple*; $\text{MV}_\text{u}$ and $\text{MV}_\rho$ refer to majority voting with uniform and optimized $\rho$, respectively.

| Model | Dataset | Bound $\text{MV}_u$ | Bound $\text{MV}_\rho$ | $\hat{A}(\text{AVG}_\rho)$ last | $\hat{A}(\text{MV}_\rho)$ last | $\hat{A}(\text{AVG}_\rho)$ all | $\hat{A}(\text{MV}_\rho)$ all |
|---|---|---|---|---|---|---|---|
| CNN LSTM | IMDB | 0.576 | 0.594 | 0.873 | 0.872 | 0.873 | 0.874 |
| ResNet20 | CIFAR-10 | 0.425 | 0.743 | 0.938 | 0.938 | 0.938 | 0.938 |
| ResNet110 | CIFAR-10 | 0.559 | 0.792 | 0.956 | 0.956 | 0.956 | 0.955 |
| WRN28-10 | CIFAR-10 | 0.794 | 0.833 | 0.967 | 0.966 | 0.967 | 0.966 |
| ResNet110 | CIFAR-100 | 0.0 | 0.127 | 0.793 | 0.791 | 0.793 | 0.791 |
| WRN28-10 | CIFAR-100 | 0.172 | 0.277 | 0.83 | 0.829 | 0.829 | 0.829 |
| ResNet50 | EyePACS | 0.686 | 0.695 | 0.911 | 0.909 | 0.912 | 0.912 |

## 5 Discussion and conclusions

Bayesian model averaging (BMA) can improve uncertainty quantification and calibration of deep neural networks (DNNs). Several lines of research study the generalization performance of the Bayes ensemble and try to improve it, where performance is measured in terms of accuracy on a test set. In general, the BMA indeed improves predictive performance *compared to a single neural network*. However, the Bayes ensemble is often – implicitly and explicitly – brought forward as a particularly good and theoretically justified approach to weight members of an ensemble to improve generalization performance. In this work, we argue conceptually in the line of Masegosa (2020), Minka (2002) and Ortega et al. (2022) and provide additional evidence that the predictive posterior is not a particularly good basis for selecting and weighting the networks in a deep ensemble. A proper Bayesian way to build a deep ensemble would be to apply Bayesian inference to the space of ensembles (Monteith et al., 2011), but this is computationally expensive and – also for that reason – is not what is typically proposed in the research directions our study addresses. A simple uniformly weighted deep ensemble can be expected to perform on par with more sophisticated BMA based approaches and it can be created more efficiently. It does not require involved sampling from the posterior and training is embarrassingly parallel.

Our proposed optimization of the weighting of DNN ensemble members using the tandem loss and additional data can improve uniform deep ensembles. Using a second-order bound to derive the optimization objective is crucial, because minimizing a first-order bound will lead to a lack of diversity similar to BMA (see also Lorenzen et al., 2019). The PAC-Bayesian weighting allows us to safely include several models from a training run, which can improve performance efficiently and make early-stopping unnecessary. It has already been argued by Sollich & Krogh (1995) that having models overfitted to a subset of the training data in a properly weighted ensemble need not be harmful. The proper weighting is important, a naïve uniform combination of many models from the same run can deteriorate performance.

In recent work, Abe et al. (2024) found that "optimization objectives which encourage or discourage predictive diversity between ensemble members" can fail to improve generalization when the ensemble members have a high complexity. In contrast to the approaches by Masegosa (2020) and Ortega et al. (2022), our suggested PAC-Bayesian bound minimization does not change the optimization of the individual ensemble members, only the final weighting is adapted. In that sense, we avoid the issues observed by Abe et al. (2024).

Although no hold-out data for early-stopping is needed, it is a limitation of the weighting approach that additional hold-out data is required. For the rather small data sets and the models considered in this study, experiments with canonical bagging showed that leaving out training data impaired the performance of the ensemble (as also observed by Lakshminarayanan et al., 2017, and Lee et al., 2015), even though bagging increases diversity. However, the data used for optimization of the weighting provide at the same time a rigorous, non-trivial upper bound on the generalization performance – and getting such formal guarantees generally requires independent data, see Gastpar et al. (2024) for theoretical arguments in the

overparameterized DNN setting. The PAC-Bayesian bounds still hold after optimization. In our case, this optimization was necessary to elevate some of the bounds from being trivial to nonvacuous guarantees on the ensemble performance. The bound presumes model combination by majority voting as prediction method, however, in our experiments we did not observe a big difference in generalization performance between majority voting and averaging predictive distributions. Thus, PAC-Bayesian optimization of the ensemble weights is highly recommended if not all data available for model construction is needed for training so that a subset can be employed for optimization and getting performance guarantees. Combining several solutions (i.e., checkpoints or snapshots) from a single training trial with PAC-Bayesian optimization presents a novel way to utilize hold-out data – for ensemble member weighting instead of checkpoint selection using early stopping.

## Code availability

Code for reproducing the experiments is submitted as supplementary material. Our work builds on several, publicly available code repositories and datasets. The IMDB training code including the CNN LSTM model was based on the Keras example, similarly to the CIFAR training code from the CIFAR-10 Keras example, including all ResNets. Both examples are distributed under the Apache 2.0 license. The Gaussian prior for the CNN LSTM on IMDB was based on the code from Wenzel et al. (2020a), also under Apache 2.0 license. The preprocessing routine of the EyePACS dataset was adapted from Google's uncertainty-baselines repository, again under the Apache 2.0 license. The Wide ResNet (Zagoruyko & Komodakis, 2016) implementation was taken from a publicly available GitHub repository. Finally, the implementation of the tandem loss PAC-Bayesian bound optimization (Masegosa et al., 2020) was taken and extended from the official implementation by the authors.

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

# A    Appendix / supplemental material

## A.1    Experimental details

The experimental setups including the hyperparameters were taken from the respective references cited and the corresponding source code. Thus, no hyperparameter tuning was performed, except for the EyePACS dataset, where the hyperparameters from the literature did not lead to convergence of the models and the learning rate was decreased by a factor of 100.

**IMDB.**    For IMDB, the described setup from Wenzel et al. (2020a) was copied as closely as possible. The dataset from the `tensorflow.keras.datasets` API was used with 20,000 word features and a maximum sequence length of 100. 20,000 training samples with 5,000 random validation samples being used for training with early-stopping due to overfitting. The test set included all 25,000 samples from the original test set. The CNN LSTM model and code were taken from the Keras example[5] and extended with a Gaussian prior $\mathcal{N}(0, I)$ for regularization, as done by the authors in their code repository[6]. As optimizer, SGD with Nesterov momentum of 0.98 and a batch size of 32 with a constant learning rate was utilized. For the cyclic learning rate in all snapshot ensemble (SSE) experiments, the original schedule from Huang et al. (2017) was used.

**CIFAR.**    The CIFAR datasets were taken from the `tensorflow.keras.datasets` API with the original train and test split. Similar to Wenzel et al. (2020a), no early-stopping with a validation set was used. The ResNet20 model was taken from the Keras example [7]. As in the example, all experiments used data augmentation during training with random left/right flipping and random cropping with $4px$ of shift horizontally and vertically. The hyperparameters for the ResNet20 were adopted from Wenzel et al. (2020a), while they were taken from Ashukha et al. (2020) for the ResNet110 and Wide ResNet28-10. As optimizer, SGD with Nesterov momentum of 0.9 and a batch size of 128 with a step-wise decreasing learning rate ($\eta$) was employed (epoch, $\eta$-multiplier): (80, 0.1), (120, 0.01), (160, 0.001), (180, 0.0005). For the ResNet110 and WRN28-10, a linearly decreasing learning rate schedule starting at half of the total number of epochs was utilized, as reported by Ashukha et al. (2020) and introduced by Garipov et al. (2018):

$$\eta(i) = \begin{cases} \eta_{\text{init}}, & i \in [0, 0.5 \cdot \text{epochs}] \\ \eta_{\text{init}} \cdot (1.0 - 0.99 \cdot (i/\text{epochs} - 0.5)/0.4), & i \in [0.5 \cdot \text{epochs}, 0.9 \cdot \text{epochs}] \\ \eta_{\text{init}} \cdot 0.01, & \text{otherwise} \end{cases} \quad (\text{A.10})$$

**EyePACS.**    The EyePACS dataset was included in the 2015 Kaggle diabetic retinopathy detection competition (Dugas et al., 2015). Diabetic retinopathy is the leading cause of blindness in the working-age population of the developed world and is estimated to affect more than 92 million people. The dataset contains high-resolution labeled RGB images of human retinas with varying degrees of diabetic retinopathy on a five-grade scale, from none (0), to mild (1), moderate (2), severe (3) and proliferating (4) development of the disease. It consists of 35126 training, 10906 validation and 42670 test images, each labeled by a medical expert. In order to binarize the labels, Band et al. (2021) follow previous work and classify moderate or worse manifestation as sight-threatening (grades 2-4), and remaining grades 0-1 as non sight-threatening. The dataset with binary labels is unbalanced, such that 19.6% of the training and 19.2% of the test set have a positive label, which is why the cross-entropy objective is weighted by the inverse of the global class distribution. Furthermore, the images exhibit different types of noise (artifacts, focus, exposure) and are expected to show label noise due to misjudgement of the medical personnel (Dugas et al., 2015). The preprocessing of images by Band et al. (2021) follows the winning entry of the original Kaggle challenge (Dugas et al., 2015). The images are first rescaled such that the retinas have a radius of 300 pixels, smoothed using local Gaussian blur with a kernel standard deviation of 100 pixels and clipped to 90% to remove boundary effects. Finally, they are resized to 512x512 pixels and stored. For our preprocessing, the implementation from Nado et al. (2021) was used[8]. Except the preprocessing, no data augmentation was used. With the

---

[5] https://github.com/keras-team/keras/blob/1a3ee8441933fc007be6b2beb47af67998d50737/examples/imdb_cnn_lstm.py

[6] https://github.com/google-research/google-research/tree/master/cold_posterior_bnn

[7] https://github.com/keras-team/keras/blob/1a3ee8441933fc007be6b2beb47af67998d50737/examples/cifar10_resnet.py

[8] https://github.com/google/uncertainty-baselines/blob/main/uncertainty_baselines/datasets/diabetic_retinopathy_dataset_utils.py

Table A.4: Hyperparameters for all experiments. *Ep.b.* (Epoch budget) refers to the experiments in section A.5.2.

| Model (Dataset) | Experiment | Val. set | $lr_{\text{init}}$ | CP | epochs | L2-reg. | LR Sched. |
|---|---|---|---|---|---|---|---|
| IMDB (CNN LSTM) | Simple | 20% | | 1 | 50 | | constant |
| | Checkp. | 20% | | 5 | 5 | | constant |
| | Bagging | Bagging | 0.1 | 1 | 50 | $\mathcal{N}(0, I)$ | constant |
| | SSE | 20% | | 5 | 50 | | cyclic |
| | Ep.b. | 20% | | 1 | variable | | constant |
| ResNet20 (CIFAR-10) | Simple | 0% | 0.1 | 5 | 200 | | step |
| | Bagging | Bagging | 0.1 | 5 | 200 | 0.002 | step |
| | SSE | 0% | 0.2 | 5 | 200 | | cyclic |
| | Ep.b. | 0% | 0.1 | 1 | variable | | step |
| ResNet110, WRN28-10 (CIFAR-10, CIFAR-100) | Simple | 0% | | 5 | 300 | | linear |
| | Bagging | Bagging | 0.1 | 5 | 300 | 0.0003 | linear |
| | SSE | 0% | | 5 | 300 | | cyclic |
| | Ep.b. | 0% | | 1 | variable | | linear |
| ResNet50 (EyePACS) | Simple | 0% | $2.3 \cdot 10^{-4}$ | 6 | 90 | $1.07 \cdot 10^{-4}$ | step |
| | Bagging | Bagging | | | | | |

optimal SGD hyperparameters from Band et al. (2021), the resulting models failed to converge in our case, which is why we reduced the learning rate and switched the optimizer to Adam. The batch size was kept at 32. For the learning rate schedule, a step-wise decrease was employed with a reduction by a factor of 0.2 at 30 and 60 epochs.

## A.2 Test-time cross-validation

Ashukha et al. (2020) describe the problem of requiring a validation set (in their case for scaling the logit outputs of neural networks with a temperature parameter), while the benchmark datasets only feature a training and test dataset (e.g. CIFAR). In that case, when splitting the training set into a training and validation subset, the performance may drop compared to methods that use the full training data because the reduced training data set is a worse description of the task. In contrast, when splitting off a validation set from the test data, one still obtains an unbiased estimate of the generalization error, but with higher variance due to smaller test dataset size. In order to reduce the variance, Ashukha et al. (2020) propose test-time cross-validation: splitting the test dataset randomly and averaging the results of the generalization estimates to reduce the variance. We follow this approach and divide the test dataset randomly in half. One half is used for the tandem loss bound optimization, while the other serves as a generalization estimate. Afterwards, the roles of the datasets are switched and the risk estimates are averaged. In our case, this setting does not result in a fair comparison. All methods should make use of the same amount of data, and algorithms that do not need a validation set could use the additional data for training, which can improve performance, in particular if data are scarce. However, one can envision a scenario where the additional data are not available during training but only later when deploying the model (e.g., local fine-tuning of centrally trained models).

## A.3 Cancellation of independent binary classification errors

Because this known result is often stated without proof, we provide an upper bound for the error probability of the majority vote for an ensemble of binary classifiers with independent errors. Its expected value was already derived in the seminal paper by Hansen & Salamon (1990).

**Theorem 3.** *Consider $M$ binary classifiers $h_i, \ldots, h_M$ mapping to $\{0, 1\}$ and the majority vote classifier given by:*

$$h_{MV}(x) = \begin{cases} 1 & \text{if } \sum_{i=1}^{M} h_i(x) \geq M/2 \\ 0 & \text{otherwise} \end{cases} \tag{A.11}$$

*We define the random variables $A_i = \mathbb{1}[h_i(x) \neq y]$ indicating a mistake by $h_i$ and assume for all $i = 1, \ldots, M$ that the risk is bounded by a constant*

$$P_{(x,y)\sim\ p}(h_i(x) \neq y) = \mathbb{E}\{A_i\} \leq p_{\max} < \frac{1}{2} \tag{A.12}$$

*and that the $A_i$ are independent. Then it holds*

$$P(h_{MV}(x) \neq y) \leq \exp\left(-\frac{2\left(\frac{M+1}{2} - Mp_{\max}\right)^2}{M}\right) \ . \tag{A.13}$$

*Proof.* Let $S_M = \sum_{i=1}^{M} A_i$. The probability $P(h_{\mathrm{MV}}(x) \neq y)$ that the majority vote classifier makes a mistake is the probability that $P(S_M > M/2)$ or alternatively $P(S_M \geq (M+1)/2)$. It holds

$$\mathbb{E}\left\{\sum_{i=1}^{M} A_i\right\} = \mathbb{E}\{S_M\} \leq Mp_{\max} \tag{A.14}$$

and we have

$$P(S_M \geq (M+1)/2) = P(S_M - \mathbb{E}\{S_M\} \geq (M+1)/2 - \mathbb{E}\{S_M\}) \tag{A.15}$$
$$\leq P(S_M - \mathbb{E}\{S_M\} \geq (M+1)/2 - Mp_{\max}). \tag{A.16}$$

With Hoeffding's inequality and $\varepsilon = (M+1)/2 - Mp_{\max}$ we get the desired bound

$$P(S_M \geq (M+1)/2) \leq P\{S_M - \mathbb{E}\{S_M\} \geq \varepsilon\} \leq \exp\left(-\frac{2\varepsilon^2}{M}\right) \tag{A.17}$$
$$= \exp\left(-\frac{2\left(\frac{M+1}{2} - Mp_{\max}\right)^2}{M}\right).$$

$\square$

### A.4 Optimization of the tandem loss bound

The following results are taken from Masegosa et al. (2020) and are just restated for completeness.

The bound in Theorem 1 is convex in $\lambda$ for a given $\rho$. Let the matrix of empirical tandem losses be defined as $\hat{L} \in \mathbb{R}^{M \times M}$ with entries $[\hat{L}]_{ij} = \hat{L}_{\mathcal{D}_{ij}}(h_i, h_j)$, where $\mathcal{D}_{ij}$ is the data used for estimating the tandem loss for hypotheses $h_i$ and $h_j$. Then the RHS of the bound defines the objective function

$$f(\rho, \lambda) = 4\left(\frac{\sum_{i,j=1}^{M} \rho_i \rho_j [\hat{L}]_{ij}}{1 - \lambda/2} + \frac{2\mathrm{KL}(\rho\|\pi) + \ln(2\sqrt{n}/\delta)}{\lambda(1 - \lambda/2)n}\right) \ . \tag{A.18}$$

From Tolstikhin & Seldin (2013) and Thiemann et al. (2017), the optimal $\lambda$ for a given $\rho$ is simply given by

$$\lambda = \frac{2}{\sqrt{\frac{2n\sum_{i,j=1}^{M}\rho_i\rho_j[\hat{L}]_{ij}}{2\mathrm{KL}(\rho\|\pi) + \ln\frac{2\sqrt{n}}{\delta}} + 1} + 1} \ . \tag{A.19}$$

Now, the gradient with respect to $\rho$ given $\lambda$ is computed. Let $[\nabla f]_i$ for denote the component of the gradient corresponding to hypothesis $h_i$ for fixed $\lambda$. Then we have

$$[\nabla f]_i = 2 \sum_{j=1}^{M} \rho(h_j)[\hat{L}]_{ij} + \frac{2}{\lambda n} \left( 1 + \ln \frac{\rho(h_i)}{\pi(h_i)} \right) \quad . \tag{A.20}$$

If $\mathcal{D}_{ij} = \mathcal{D}_{kl}$ for $i, j, k, l \in \{1, \dots, M\}$, then the loss is convex in $\rho$ for a given $\lambda$. This is true in our experiments with a separate data set for tuning the ensemble, but not when using bagging. We refer to Masegosa et al. (2020) for details.

In practice, for a given ensemble, the bound is optimized in an iterative procedure. Starting from a uniform $\rho^{(0)}$, in iteration $t = 0, 1, \dots$ one computes $\lambda^{(t)}$ using $\rho^{(t)}$ and equation (A.19) and then $\rho^{(t+1)}$ by performing a gradient-based optimization step using $\lambda^{(t)}$ and equation (A.20), see Algorithm 1 for the algorithm in pseudo-code. The optimization problem is low-dimensional ($\rho \in \mathbb{R}^M$) and converges quickly. We employed iRprop (Igel & Hüsken, 2003) for gradient-based optimization as done by Wu et al. (2021).

---

**Algorithm 1:** Iterative optimization of the $\lambda$-bound

**Data:** Matrix of empirical tandem losses $\hat{L}$, stopping criterion parameter $T_\rho$
**Result:** Optimized $\rho$ and $\lambda$
$\rho \leftarrow \frac{1}{M}\mathbf{1}_M$               `// 1`$_M$` is the `$M$`-dimensional vector of all ones`
$f(\rho, \lambda) = 4 \left( \frac{\sum_{i,j=1}^{M} \rho_i \rho_j [\hat{L}]_{ij}}{1 - \lambda/2} + \frac{2\mathrm{KL}(\rho\|\pi) + \ln(2\sqrt{n}/\delta)}{\lambda(1 - \lambda/2)n} \right)$     `// Definition of objective function`
**while** *objective function $f$ improves after $\lambda$ update* **do**
    `// Update `$\lambda$` using current `$\rho$
    $\lambda \leftarrow 2 \left( \sqrt{\frac{2n \sum_{i,j=1}^{M} \rho_i \rho_j [\hat{L}]_{ij}}{2\mathrm{KL}(\rho\|\pi) + \ln \frac{2\sqrt{n}}{\delta}} + 1} + 1 \right)^{-1}$
    `// Update `$\rho$` using current `$\lambda$
    **repeat**
        **for** $i = 1$ **to** $M$ **do**
            $[\nabla f]_i \leftarrow 2 \sum_{j=1}^{M} \rho(h_j)[\hat{L}]_{ij} + \frac{2}{\lambda n} \left( 1 + \ln \frac{\rho(h_i)}{\pi(h_i)} \right)$
        Update $\rho$ by an iteration of gradient-based optimization (e.g., iRprop) using $\nabla f$
    **until** *objective function $f$ does not improve for $T_\rho$ steps*

---

## A.5 Further experimental results

### A.5.1 Bagging

In the bagging experiments, we used different training data for each ensemble member to increase diversity. We drew bootstrap samples from the training data uniformly at random with replacement. The sample size was equal to the size of the training data. The experiments illustrate the effect of decreasing the training data volume (e.g., to use the data for bound optimization).

The individual network performance suffered from omitting unique training data points, and although the networks are assumed to be more diverse, the resulting ensembles performed worse performance across all experiments. The results are visualized in Figure A.7, which shows that none of the ensembles, neither uniform nor optimized in their weighting, could match the Bayesian reference. This is in line with the literature (Lakshminarayanan et al., 2017; Lee et al., 2015), which finds the same decrease in performance for neural network ensembles. PAC-Bayesian optimization is therefore best suited for large datasets with additional data that is not essential for training. On a positive note, random initialization and stochastic training appears to be enough randomization for creating diverse ensembles.

Table A.5: Test accuracies for $\mathrm{AVG}_\rho$ and $\mathrm{MV}_\rho$ and PAC-Bayesian bounds for all experiments. The bounds were computed for the *all* setting, except for IMDB, where checkpoints were taken in a separate experiment due to early-stopping in *Simple*; $\mathrm{MV}_u$ and $\mathrm{MV}_\rho$ refer majority voting with uniform and optimized $\rho$, respectively.

| Model (Dataset) | Experiment | Bound $\mathrm{MV}_u$ | Bound $\mathrm{MV}_\rho$ | $\hat{A}(\mathrm{AVG}_\rho)$ last | $\hat{A}(\mathrm{MV}_\rho)$ last | $\hat{A}(\mathrm{AVG}_\rho)$ all | $\hat{A}(\mathrm{MV}_\rho)$ all |
|---|---|---|---|---|---|---|---|
| CNN LSTM (IMDB) | Simple | 0.585 | 0.59 | 0.872 | 0.873 | - | - |
| | Checkpointing | 0.576 | 0.594 | 0.873 | 0.872 | 0.873 | **0.874** |
| | Bagging | 0.579 | 0.585 | 0.869 | 0.867 | - | - |
| | SSE | 0.459 | 0.575 | 0.741 | 0.74 | **0.874** | 0.872 |
| | Ep.b. (500) | 0.571 | 0.574 | 0.873 | 0.872 | - | - |
| ResNet20 (CIFAR-10) | Simple | 0.425 | 0.743 | **0.938** | **0.938** | **0.938** | **0.938** |
| | Bagging | 0.387 | 0.691 | 0.929 | 0.928 | 0.929 | 0.928 |
| | SSE | 0.593 | 0.612 | 0.894 | 0.893 | 0.908 | 0.909 |
| | Ep.b. (1500) | 0.718 | 0.718 | 0.937 | 0.934 | - | - |
| ResNet110 (CIFAR-10) | Simple | 0.559 | 0.792 | **0.956** | **0.956** | **0.956** | 0.955 |
| | Bagging | 0.555 | 0.763 | 0.949 | 0.948 | 0.949 | 0.948 |
| | SSE | 0.785 | 0.802 | 0.955 | 0.954 | 0.954 | 0.953 |
| | Ep.b. (1500) | 0.799 | 0.8 | 0.954 | 0.953 | - | - |
| | Ep.b. (300) | 0.735 | 0.735 | 0.942 | 0.939 | - | - |
| Wide ResNet28-10 (CIFAR-10) | Simple | 0.794 | 0.833 | **0.967** | 0.966 | **0.967** | 0.966 |
| | Bagging | 0.764 | 0.808 | 0.959 | 0.958 | 0.959 | 0.958 |
| | SSE | 0.843 | 0.848 | 0.963 | 0.964 | 0.964 | 0.964 |
| | Ep.b. (1500) | 0.832 | 0.834 | 0.966 | 0.965 | - | - |
| | Ep.b. (300) | 0.805 | 0.806 | 0.961 | 0.957 | - | - |
| ResNet110 (CIFAR-100) | Simple | 0.0 | 0.127 | 0.793 | 0.791 | 0.793 | 0.791 |
| | Bagging | 0.0 | 0.0 | 0.771 | 0.769 | 0.771 | 0.768 |
| | SSE | 0.083 | 0.125 | 0.79 | 0.787 | **0.794** | 0.793 |
| | Ep.b. (1500) | 0.091 | 0.099 | 0.783 | 0.777 | - | - |
| | Ep.b. (300) | 0.0 | 0.0 | 0.749 | 0.728 | - | - |
| Wide ResNet28-10 (CIFAR-100) | Simple | 0.172 | 0.277 | **0.83** | 0.829 | 0.829 | 0.829 |
| | Bagging | 0.041 | 0.173 | 0.812 | 0.811 | 0.812 | 0.811 |
| | SSE | 0.276 | 0.289 | 0.818 | 0.818 | 0.826 | 0.827 |
| | Ep.b. (1500) | 0.271 | 0.276 | 0.829 | 0.822 | - | - |
| | Ep.b. (300) | 0.233 | 0.233 | 0.815 | 0.811 | - | - |
| ResNet50 (EyePACS) | Simple | 0.686 | 0.695 | 0.911 | 0.909 | **0.912** | **0.912** |
| | Bagging | 0.613 | 0.638 | 0.894 | 0.893 | 0.896 | 0.895 |

### A.5.2 Training time comparison

It is not straight-forward to compare the computational resources required by the different approaches due to their different training and inference procedures. The BMA based approaches require training of the neural network and sampling the posterior, where the latter can be time consuming. In contrast, minimization of the PAC-Bayesian bound is highly efficient and its computational costs can be neglected. Simple deep ensembles are embarrassingly parallel, while methods that consider ensemble members from one process cannot be fully parallelized.

But even for simple ensembles, there can be a trade-off between number of networks and the number of training iterations spent on each network. To explore this trade-off, we considered the worst case scenario for deep ensemble training and assume that there is no parallelization. Furthermore, we assumed that the sampling from the posterior does not take any time. That is, for methods based on BMA only the training

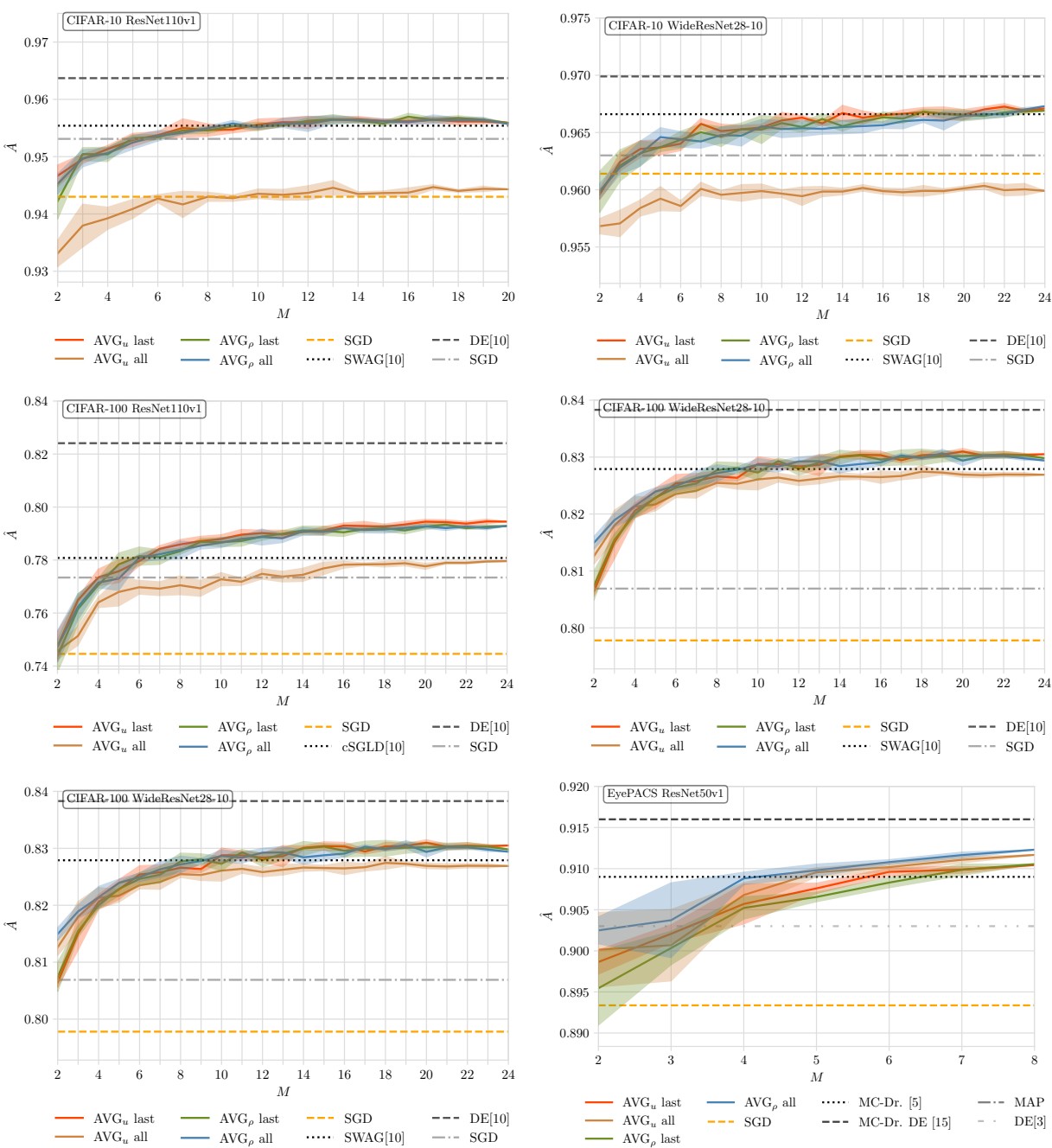

Figure A.5: Mean test accuracy $\hat{A} \pm \sigma$ over five ensembles for uniformly and PAC-Bayesian weighted deep ensembles ($\text{AVG}_u$ and $\text{AVG}_\rho$), using only the last or all training checkpoints, and best single model (SGD). References are Bayesian ensembles cSGHMC-ap (Wenzel et al., 2020a), cSGLD (Ashukha et al., 2020), *MC-Dr*opout and *MC-Dr*opout *D*eep *E*nsemble (i.e., ensemble of Bayesian ensembles), as well as simple *D*eep *E*nsembles for EyePACS from Band et al. (2021). Numbers in brackets indicate the ensemble sizes.

epochs for optimizing the network are counted, not the sampling. In this sequential setting, we asked, given an overall budget of $B$ training epochs, how does the performance of our deep ensembles depend on the number $M$ of networks when each network is trained for $\lfloor B/M \rfloor$ epochs. For the experiments taken from Wenzel et al. (2020a), we set $B = 1500$ for CIFAR-10 and $B = 500$ for IMDB, corresponding to the number

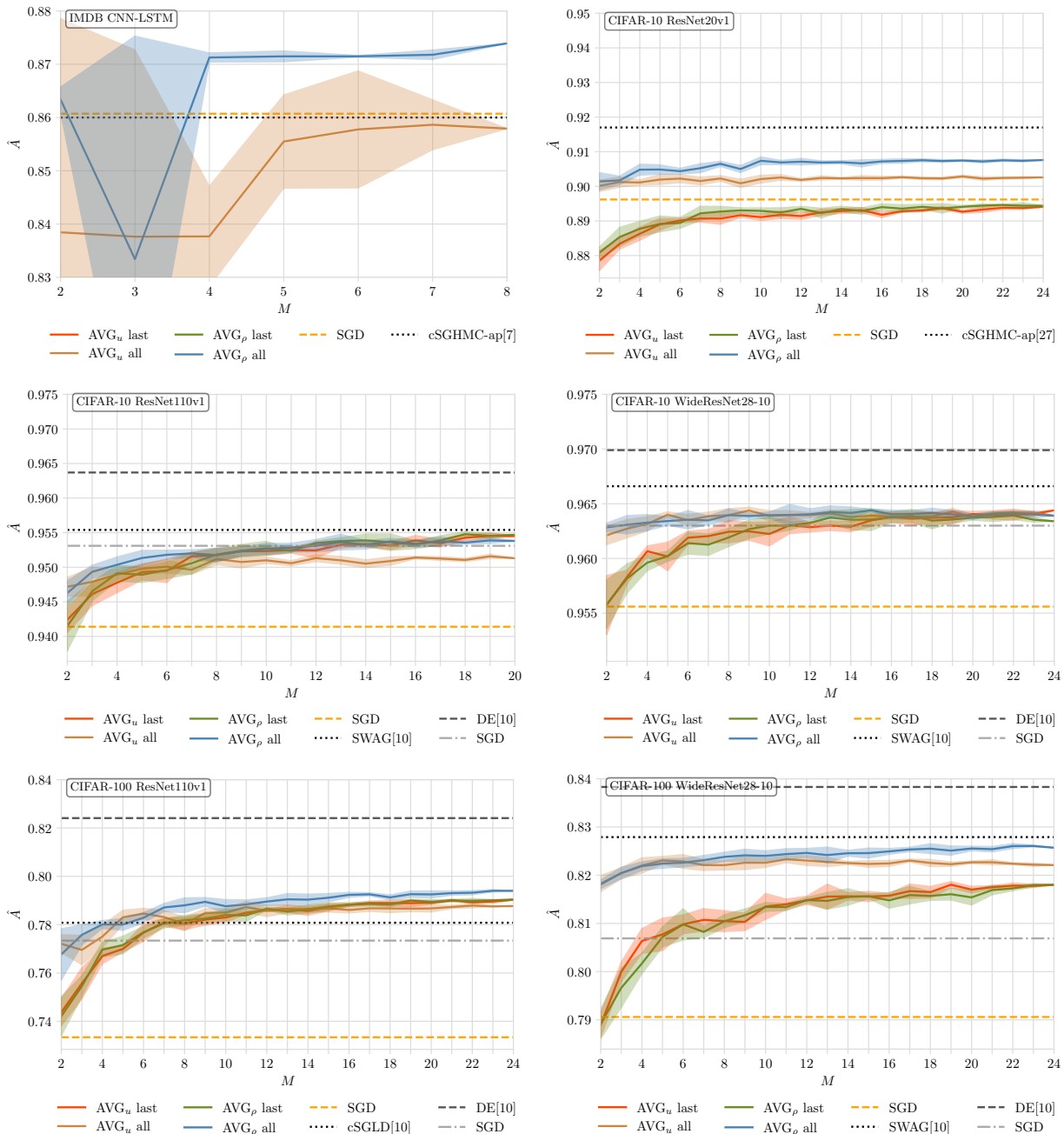

Figure A.6: Snapshot ensemble (SSE) experiments: Mean ensemble accuracy (5 ensembles, $\pm\sigma$)

of training epochs used for the Bayes ensemble. The results are shown in Figure A.8. Even in this biased comparison, there were choices of $M$ for which the simple deep ensemble outperformed the Bayesian approach.

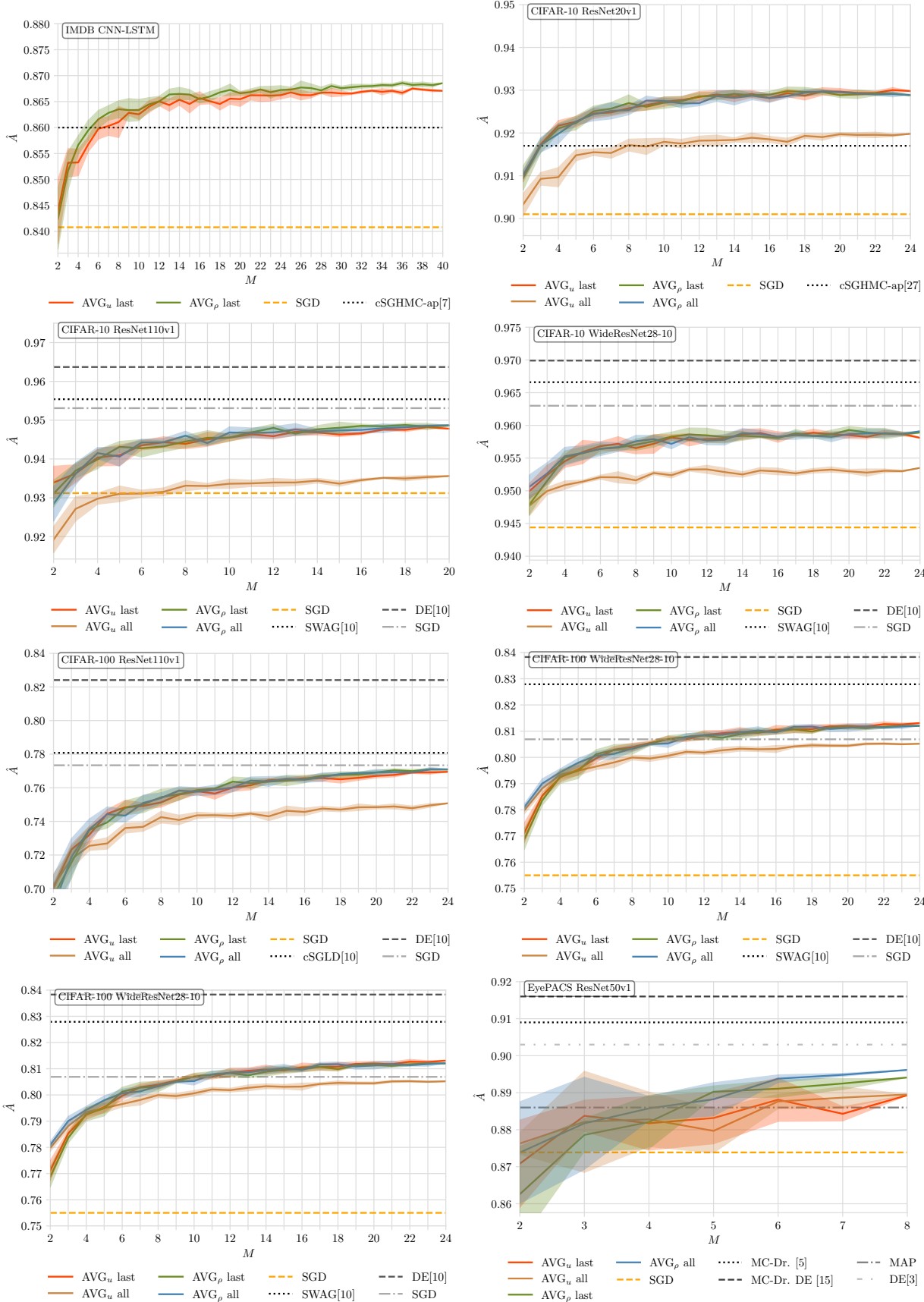

Figure A.7: Bagging: Mean ensemble accuracy (5 runs, $\pm\sigma$ highlighted)

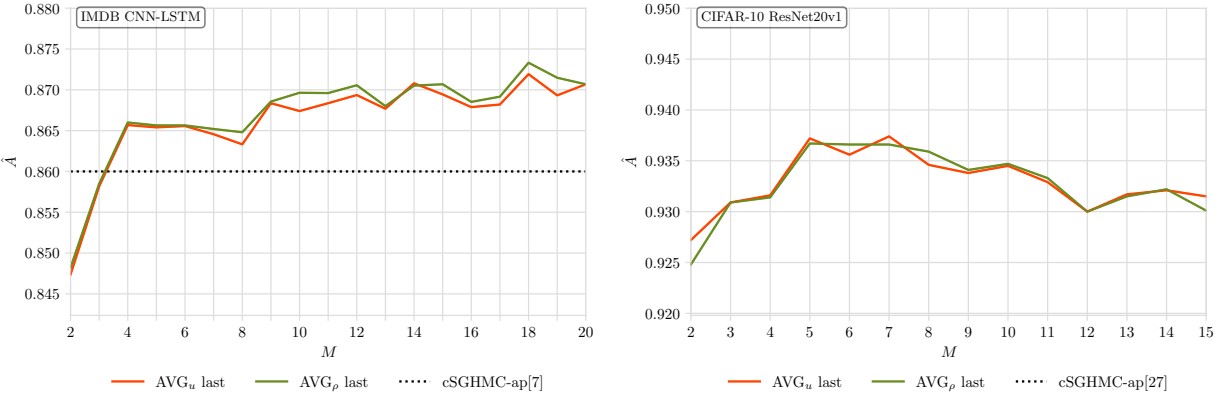

Figure A.8: Epoch budget in a sequential training setting: Mean ensemble accuracy (5 ensembles, $\pm\sigma$). The number of epochs is given by $\lfloor B/M \rfloor$, where $b = 1500$.

