# OpenReview forum: "On Uniform, Bayesian, and PAC-Bayesian Deep Ensembles}"
_TMLR — Rejected by TMLR_

### Review · Reviewer_GcA5 · 2025-02-03

**Summary Of Contributions:**

In this paper, the authors argue that viewing deep ensembles from the Bayesian perspective, i.e. from the p.o.v. of Bayesian model averaging (BMA), is flawed if we want to increase generalization performance. This is because, in the infinite-data limit, the posterior distribution of a Bayesian model collapses to a MAP (point) estimate and so, there is no diversity in the samples that are being used for computing the BMA prediction. Therefore, the reasoning behind deep ensembles, i.e. cancellation of errors due to averaging does not apply anymore.

The authors thus argue that the proper way to "be Bayesian" about deep ensembles is to be Bayesian (put priors etc.) on the space of ensemble models. This way, in the infinite-data limit, the posterior will collapse to the best *ensemble* model, not to a single model like in Bayesian neural networks.

However, the aforementioned approach can be expensive. Hence, the authors propose to use the tandem loss found in PAC-Bayesian bound literature.

**Audience:**

Yes

**Claims And Evidence:**

Yes

**Requested Changes:**

- Structure the discussion in the text more, and make differences in terminologies (like "Bayes ensemble" above) clearer.
- Show and highlight (e.g. in a separate section), how can one perform the proposed optimization in practice.
- Make the discussion about experimental results clearer.

**Strengths And Weaknesses:**

**Strengths**

- The discussion regarding cancellation of errors, BMA, deep ensemble, and a proper Bayesian way of doing ensemble learning is interesting. It provides an alternative view of deep ensembles from the Bayesian perspective: one can see it from the BMA angle, or from the Bayesian inference in the space of ensemble models.
- The experimental validations are done with large neural nets (ResNets 20, 101 etc) which I appreciate.

**Weaknesses**

- The text is hard to follow.
    - For example, in Sec. 2.1, "Bayes ensemble" equals BMA, but in other places (e.g. Abstract and Intro), it also means weighting the BMA with the posterior probability.
    - Sec. 3.2. is quite hard to parse. It can be improved by breaking the section down into more paragraphs.
    - etc.
- The proposed method (optimization with the tandem loss), which is the main methodological contribution of this paper, is not described well. After several passes of reading, I'm still not very sure how one could perform the optimization in practice.
- The experimental results are also hard to parse.
   - E.g. it's very hard to differentiate the colors in Fig. 1. And even then, the results don't look significant.
   - E.g. in the tables, the difference between the numbers seems minuscule.
   - etc.

---

### Review · Reviewer_CmGw · 2025-02-12

**Summary Of Contributions:**

- This work presents comparisons between three conceptually different ways to combine predictive models into an ensemble: *Uniform* ensembles, in which independently trained ensemble members are weighted equally, *Bayesian* ensembles, in which ensemble members are thought of as samples from some posterior distribution over possible models, and *PAC-Bayesian* ensembles, which are formed by optimizing PAC-Bayes generalization bounds over a weighted collection of classifiers. In this particular work, the authors study a method which optimizes solely the weighting of ensemble members (as opposed to additionally optimizing network parameters themselves), admitting valid bounds on in distribution generalization performance. The work focuses on the setting of classification tasks in which ensemble members are deep neural networks, and discuss results using either averaging or voting schemes to combine neural network predictions.
- In sections 2 and 3, the authors discuss a variety of conceptual differences between different methods of forming ensembles, with a particular focus on the performance of Bayesian ensembles, and reasons why they may be suboptimal for the purpose of improving predictive performance. They formalize arguments that are often cited in the literature regarding 1) the reasons why ensembles may improve predictive performance and 2) why Bayesian ensembles may not be well-suited to take advantage of these potential benefits. They then argue that the second order PAC-Bayes bound they will optimize is a conceptually better suited candidate to take advantage of the properties of ensemble members. They finally discuss potential benefits of their approach to cases where we have access to intermediate checkpoints from model training, and PAC-Bayes bounds can efficiently select models to upweight in the resulting ensemble.
- Section 4 presents experimental results, in which the authors compare a variety of uniform, Bayesian, and PAC-Bayesian ensembles across 4 experimental datasets, using the same component model type within each dataset for comparison. They generally compare one Bayes ensemble per-experimental dataset to the performance of a family of uniform and PAC-Bayesian ensembles, sweeping over the number of component models they consider. PAC-Bayes and uniform ensembles seem quite similar in terms of depicted performance, and we can often find some range of component model number which leads both PAC-Bayes and uniform ensembles to outperform the Bayesian baseline. This result seems to apply regardless of if we use best checkpoints from independently trained models as potential ensemble members, or include checkpoints from intermediate training with/without a cyclic learning rate (checkpoint vs SSE ensembles). Finally, they show some representative examples of the ensemble member weighting learned by optimizing either (I believe) first order or second order PAC-Bayes bounds, and show PAC-Bayes bounds for performance on a subset of the ensembles that they considered.

**Audience:**

Yes

**Claims And Evidence:**

No

**Requested Changes:**

**Comparison to baselines**
- 1A: In Figure 1, I'm not sure it makes much sense to make comparisons to a single Bayesian baseline, given that the range of ensemble members for uniform and PAC-Bayesian ensembles in the main result is seemingly arbitrary. Is there any basis for the choice of ensemble member count? One could imagine trying to normalize to different quantities (the simplest of being number of ensemble members, but one could imagine parameters, flops, or epochs as is done in the appendix), but I do not think it makes much sense to claim that uniform/PAC Bayesian ensembles can outperform bayesian ensembles given "enough" compute, as it were. If we were to control for the number of deep ensemble members/epochs, the superiority of Uniform/PAC-Bayesian methods to Bayesian ensembles seems questionable (Figure 1 and Appendix A.5.2). This comment also applies to Table 1. We can imagine that the performance of any of these methods has some meaningful dependence upon the number of included ensemble members (plausibly improving, given the results in Figure 1), and it seems that there are generally far more ensemble members in the non-Bayesian ensembles than in the Bayesian ones. The ambiguity here makes it difficult to evaluate to what degree the results presented support either claim 1 or claim 2, and addressing these concerns would be critical for me reconsider my recommendation for this paper.
- 1B: Allowing PAC-Bayesian ensembles access to additional validation data does not facilitate fair comparisons to Uniform or Bayesian methods. I believe a much fairer comparison would be either 1) Drawing the validation set for PAC-Bayes bound optimization from the training data, or 2) Allowing other ensemble members to train on the validation split used to optimize the PAC-Bayes bound. There are other methods which
- At the moment it is very difficult to evaluate the degree to which claim 1 or 2 are supported. Addressing these concerns would be critical for me reconsider my recommendation for this paper.

I will note that the authors seem to some degree aware of these issues- experiments in A.5.1 and A.5.2 are referenced as controlling for differences in training dataset size, or controlling the number of overall training epochs. I found the results in these figures to be inconclusive as regarding my conerns above. In particular, 1) bagging has no clear relationship to the difference in training dataset size between uniform and PAC-Bayes ensembles and 2) the results when controlling for training epochs show some definitive loss in performance that makes it even more difficult to understand the results presented in the main text.

**Understanding diversity of PAC-Bayes Ensemble predictions.**
How does optimizing the second order PAC-Bayes bound actually influence the diversity of ensemble member predictions? Looking at Table 2, it seems like there are some cases where the bound for uniform weighting and the bound for PAC-Bayes optimized weighing are quite close. Can you show us the actual values of the tandem loss for these different weightings? Other common measures of ensemble diversity as discussed in Kuncheva and Whitaker (2003) would also be interesting. Additionally, how does the choice of $\rho$ weighting affect the weighted average performance of ensemble members, relative to the uniform baseline?


**General Clarity**
- Figure 1 is generally confusing.
	- The colors are very difficult to distinguish from one another.
	- The meaning of the acronym "DE" is not described for CIFAR100 in the caption.
	- It would be useful if there was a consistent visual way of distinguishing bayesian vs. uniform baselines that you consider.
- It would be very useful to know more details about about the Bayesian baselines you consider:
    -  Please provide more details about SGHMC-ap. Does this method correspond to sg-MCMC in Wenzel et al. 2020a? Or rather, does it implement some form of rejection sampling? This change in nomenclature is confusing.
    -  Please provide more details about how the bayesian baselines you are comparing to in the other datasets (cSGLD and MC Dropout). Not all Bayesian ensembling methods are equal and it would be useful to know more about the specific methods you are considering as baselines.
- It is not clear what bound is being optimized in the second rows of Figure 2 and 3. Is this a first order PAC-Bayes bound optimization? Please describe the experimental setup this figure more carefully.
- For optimization purposes, what is the prior $\pi$?

## Small points
- Shouldn't $\rho$ be a member of the $C$ class probability simplex, not $R^N$?
- In Theorem 1, the bracket directions are inverted for $\lambda$.
- I believe the bound in equation 3 is useful for intuition, but I think it is worth citing that Hansen and Salamon provide an explicit calculation of this probability in the general multiclass case as equation 8.

## References
- Hansen, Lars Kai, and Peter Salamon. "Neural network ensembles." IEEE transactions on pattern analysis and machine intelligence 12.10 (1990): 993-1001.
- Kuncheva, Ludmila I., and Christopher J. Whitaker. "Measures of diversity in classifier ensembles and their relationship with the ensemble accuracy." Machine learning 51 (2003): 181-207.

**Strengths And Weaknesses:**

## Strengths
- The authors present a thoughtful and didactic consolidation of literature around different methods of forming ensembles from component models.
- The optimization of ensemble member weighting that they study is to my knowledge novel, as well as the presentation of bounds on deep ensemble performance.
## Weaknesses
I have grouped the weaknesses I describe into broad categories. Each group corresponds to a section of the requested changes that follow.
1. **Comparison to baselines.** There are two major issues in the experimental comparisons presented that complicate the interpretation of results presented.
	- A) The first is standardization in the number of ensemble members which are considered in the comparison between ensemble members. Across Uniform, Bayesian, and PAC-Bayesian ensembles, there is too little discussion in the main text as to what considerations were used to determine 1) the number of ensemble members included in the Bayesian ensembles. 2) The range of ensemble member counts in corresponding uniform and PAC-Bayesian cases, and why this range should constitute a fair comparison This is especially the case in Figure 1 and to a lesser extent in Table 1 as well.
	- B) The second is the amount of data that is available to each ensemble during training. The authors note that they use test-time cross validation approach to fit the weights of PAC-Bayes ensembles, effectively making more data available to the PAC-Bayes ensembles than to either the Bayesian baselines or the uniformly weighted ensembles. I consider this a significant weakness of the work that makes the conclusions harder to evaluate.

2. **Understanding diversity of PAC-Bayes Ensemble predictions.** The method presented in this paper learns a weighting of ensemble members based on minimization of a loss that reduces the pairwise correlation in errors between ensemble members. Thus, improvements to bounds in ensemble performance are implicitly targeting increased diversity in the predictions of resulting ensemble members. Given that the performance of PAC-Bayesian ensembles is quite similar to uniform ensembles, it would be useful to know how optimization of the PAC-Bayes bound actually influences the diversity of ensemble predictions. Additionally, recent work (Kobayashi et al. 2021, Thiesen et al. 2023, Abe et al. 2024) shows that high-capacity models (such as many of the model architectures studied here) generate predictions which are quite similar, reducing the potential benefits of ensembling. Furthermore, efforts to increase diversity in these models reduces the performance of individual models, leading to overall worse ensemble performance. It would therefore be useful to understand if there is a similar tradeoff (i.e. prioritizing different errors leads to worse component models) when selecting a weighting $\rho$.

3. **General Clarity**
- In general, I found the presentation of this paper quite difficult to follow. While interesting, I felt that there was considerable overlap in the material presented in sections 2 and 3 (e.g. both discuss Bayesian ensembling and the cancellation of errors with strong connections to previous work; likewise PAC-Bayesian ensembling.)
- I think that more care should be taken to describe experimental results. The figures and tables presented are quite dense, and often it is quite difficult to follow which features of the plots correspond to the results discussed in the main text.

## References
- Kobayashi, Seijin, Johannes von Oswald, and Benjamin F. Grewe. "On the reversed bias-variance tradeoff in deep ensembles." ICML, 2021.
- Theisen, Ryan, et al. "When are ensembles really effective?." Advances in Neural Information Processing Systems 36 (2024).
- Abe, Taiga, et al. "Pathologies of Predictive Diversity in Deep Ensembles." Transactions on Machine Learning Research.

---

### Review · Reviewer_47Hq · 2025-02-27

**Summary Of Contributions:**

This paper has investigated different variants of deep ensembles, including the uniform deep ensemble, Bayesian deep ensemble, and PAC-Bayesian deep ensemble. The main contributions of this paper are the three claims made in Section 3, and the experiment tests of these claims in Section 4. The three claims are:

1) The Bayes ensemble is not a particularly good way to select and weight networks in a deep ensemble.

2) PAC-Bayesian weighting optimized using the tandem loss can improve the generalization performance of a deep ensemble.

3) Optimizing the weighting using the tandem loss allows inclusion of several models from a training run in a way that efficiently improves performance and makes early-stopping unnecessary

Extensive experiment results are provided in Section 4.

**Audience:**

Yes

**Claims And Evidence:**

Yes

**Requested Changes:**

Please try to address the weaknesses listed above.

**Strengths And Weaknesses:**

**Strengths:**

1) This paper aims to enhance the community's understanding of a key problem in ensemble learning.

2) Extensive experiment results have been provided, and the code is also available.

3) The authors have done a good job to review literature and synthesize the existing results.

**Weaknesses:**

1) The writing of this paper needs to be further improved, especially for Section 3. Specifically, in the technical discussion in Section 3.1 and 3.2, if possible, please use clean and rigorous math notations to present the ideas. I have found some discussion in this section unclear and non-rigorous. I am relatively familiar with this area, but still find parts of this section hard to digest. My understanding is that writing is particularly important for this paper since researchers with different backgrounds might have different perspectives about deep ensembles. Improved writing will make it much easier for researchers to digest the perspectives and results of this paper.

2) This paper summarizes its main perspectives in three "claims", which are stated in a non-rigorous way (e.g. what do you mean by "not a particularly good way" in Claim 1?). I strongly recommend the authors to rewrite the "claims" in a rigorous way. Ideally, the authors might also use hypothesis tests to formally test their claims: the authors might state their counterclaims (i.e. the opposites of their claims) as null hypotheses, and try to formally reject them.

---

### Decision · Action_Editor_RW3m · 2025-04-04

**Recommendation:** Reject

**Comment:**

Beyond the issues around claims and evidence, many reviewers noted issues with clarity in the paper. The method names and notation used in the tables and figures are hard to parse and generally do not match the terms used in the rest of the text. The authors' use of the term "claim" is perhaps misleading, as often what is claimed is an empirical observation rather than a provable mathematical fact. While the authors made some effort towards improving clarity, I would recommend a significant effort to (1) clearly explain methods, (2) use consistent terminology and notation, and (3) reduce the verbosity in the main text.

**Audience:**

Deep ensembles and Bayesian neural networks are topics of interest to the TMLR community. A comparison of the two methods, as well as a PAC Bayesian weighting scheme of ensemble members, will be of interest to the TMLR community.

**Claims And Evidence:**

After reading the reviews, rebuttals, and discussion, there are outstanding issues that make me question the findings in the experimental results of this paper.

(1) The comparison against BMA ensembles of fixed size is unfair and perhaps misleading. It is not surprising that an arbitrarily large standard deep ensemble can outperform a BMA of a fixed size.

(2) The comparison between simple and PAC-Bayes-weighted ensembles is also unfair, since the PAC-Bayes-weighted ensembles have access to validation data that the simple ensembles do not have access to.

While the authors provided an argument for these decisions in the revision, I believe that proper comparisons that mitigate these difference are necessary to support their claims.

**Resubmission Of Major Revision:**

The authors may consider submitting a major revision at a later time.